# EARLY STOPPING FOR DEEP IMAGE PRIOR

## ABSTRACT

Deep image prior (DIP) and its variants have shown remarkable potential for solving inverse problems in computational imaging (CI), *needing no separate training data*. Practical DIP models are often substantially overparameterized. During the learning process, these models first learn the desired visual content and then pick up the potential modeling and observational noise, i.e., overfitting. Thus, the practicality of DIP hinges on early stopping (ES) that can capture the transition period. In this regard, most previous DIP works for CI tasks only demonstrate the potential of the models—reporting the peak performance against the groundtruth but providing no clue about how to operationally obtain near-peak performance *without access to the groundtruth*. In this paper, we set to break this practicality barrier of DIP, and propose an efficient ES strategy that consistently detects near-peak performance across several CI tasks and DIP variants. Simply based on the running variance of DIP intermediate reconstructions, our ES method not only outpaces the existing ones—which only work in very narrow regimes, but also remains effective when combined with methods that try to mitigate overfitting.

## 1 INTRODUCTION

Inverse problems (IPs) are prevalent in computational imaging (CI), ranging from basic image denoising, super-resolution, and deblurring, to advanced 3D reconstruction and major tasks in scientific and medical imaging (Szeliski, 2022). Despite the disparate settings, all these problems take the form of recovering a visual object $\boldsymbol{x}$ from $\boldsymbol{y} = f(\boldsymbol{x})$, where $f$ models the forward process to obtain the observation $\boldsymbol{y}$. Typically, these visual IPs are underdetermined: $\boldsymbol{x}$ cannot be uniquely determined from $\boldsymbol{y}$. This is exacerbated by potential modeling (e.g., linear $f$ to approximate a nonlinear process) and observational (e.g., Gaussian or shot) noise, i.e., $\boldsymbol{y} \approx f(\boldsymbol{x})$. To overcome the nonuniqueness and improve noise stability, people often encode a variety of problem-specific priors on $\boldsymbol{x}$ when formulating IPs. Traditionally, IPs are phrased as regularized data-fitting problems:

$$\min_{\boldsymbol{x}} \; \ell(\boldsymbol{y}, f(\boldsymbol{x})) + \lambda R(\boldsymbol{x}) \qquad \ell(\boldsymbol{y}, f(\boldsymbol{x})) : \text{data-fitting loss}, \; R(\boldsymbol{x}) : \text{regularizer} \qquad (1)$$

where $\lambda$ is the regularization parameter. Here, the loss $\ell$ is often chosen according to the noise model, and the regularizer $R$ encodes priors on $\boldsymbol{x}$. The advent of deep learning (DL) has revolutionized how IPs are solved: on the radical side, deep neural networks (DNNs) are trained to directly map any given $\boldsymbol{y}$ to an $\boldsymbol{x}$; on the mild side, pretrained or trainable DL models are taken to replace certain nonlinear mappings in numerical algorithms for solving Eq. (1) (e.g., plug-and-play, and algorithm unrolling). Recent surveys Ongie et al. (2020); Janai et al. (2020) on these developments trust large training sets $\{(\boldsymbol{y}_i, \boldsymbol{x}_i)\}$ to adequately represent the underlying priors and/or noise distributions. **This paper concerns another family of striking ideas that require no separate training data**.

**Deep image prior (DIP)**    Ulyanov et al. (2018) proposes parameterizing $\boldsymbol{x}$ as $\boldsymbol{x} = G_{\boldsymbol{\theta}}(\boldsymbol{z})$, where $G_{\boldsymbol{\theta}}$ is a trainable DNN parametrized by $\boldsymbol{\theta}$ and $\boldsymbol{z}$ is a trainable or frozen random seed. **No separate training data other than $\boldsymbol{y}$ are used!** Putting the reparametrization into Eq. (1), we obtain

$$\min_{\boldsymbol{\theta}} \; \ell(\boldsymbol{y}, f \circ G_{\boldsymbol{\theta}}(\boldsymbol{z})) + \lambda R \circ G_{\boldsymbol{\theta}}(\boldsymbol{z}). \qquad (2)$$

$G_{\boldsymbol{\theta}}$ is often "overparameterized"—containing substantially more parameters than the size of $\boldsymbol{x}$, and "structured"—e.g., consisting of convolution networks to encode structural priors in natural visual objects. The resulting optimization problem is solved via standard first-order methods for modern DL (e.g., (adaptive) gradient descent). When $\boldsymbol{x}$ has multiple components with different physical

meanings, one can naturally parametrize $x$ using multiple DNNs. This simple idea has led to surprisingly competitive results on numerous visual IPs, from low-level image denoising, super-resolution, inpainting (Ulyanov et al., 2018; Heckel & Hand, 2019; Liu et al., 2019) and blind deconvolution (Ren et al., 2020; Wang et al., 2019; Asim et al., 2020; Tran et al., 2021; Zhuang et al., 2022a), to mid-level image decomposition and fusion (Gandelsman et al., 2019; Ma et al., 2021), and to advanced CI problems (Darestani & Heckel, 2021; Hand et al., 2018; Williams et al., 2019; Yoo et al., 2021; Baguer et al., 2020; Cascarano et al., 2021; Hashimoto & Ote, 2021; Gong et al., 2022; Veen et al., 2018; Tayal et al., 2021; Zhuang et al., 2022b); see the survey Qayyum et al. (2021).

**Overfitting issue in DIP**    A critical detail that we have glossed over is **overfitting**. Since $G_\theta$ is substantially overparameterized, $G_\theta(z)$ can represent arbitrary elements in the $x$ domain. Global optimization of (2) would normally lead to $y = f(G_\theta(z))$, but $G_\theta(z)$ may not reproduce $x$, e.g., when $f$ is non-injective, or $y \approx f(x)$ so that $G_\theta(z)$ also accounts for the modeling and observational noise. Fortunately, DIP models and first-order optimization methods together offer a blessing: in practice, $G_\theta(z)$ has a bias toward the desired visual content and learns it much faster than learning noise. So the reconstruction quality climbs to a peak before potential degradation due to noise; see Fig. 1. This "early-learning-then-overfitting" (ELTO) phenomenon has been repeatedly reported in

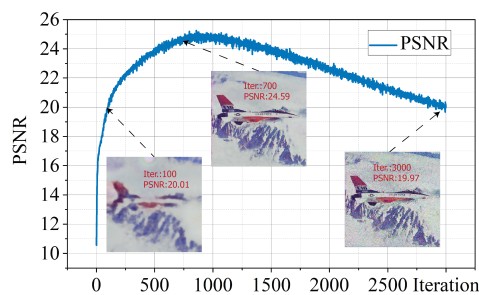

Figure 1: The "early-learning-then-overfitting" (ELTO) phenomenon in DIP for image denoising. The quality of the estimated image climbs to a peak first and then plunges once the noise is picked up by the model $G_\theta(z)$ also.

prior works and is also backed by theories on simple $G_\theta$ and linear $f$ (Heckel & Soltanolkotabi, 2020b;a). The successes of DIP models claimed above are mostly conditioned on that appropriate **early stopping** (ES) around the performance peaks can be made.

**Is ES for DIP trivial?**    Natural ideas trying to perform good ES can fail quickly. **(1) Visual inspection**: This subjective approach is fine for small-scale tasks involving few problem instances, but quickly becomes infeasible for many scenarios, such as (a) large-scale batch processing, (b) recovery of visual contents tricky to be visualized and/or examined by eyes (e.g., 3D or 4D visual objects), and (c) scientific imaging of unfamiliar objects (e.g., MRI imaging of rare tumors, and microscopic imaging of new virus species); **(2) Tracking full-reference/no-reference image quality metrics (FR/NR-IQMs)**: Without the groundtruth $x$, computing any FR-IQM and hence tracking their trajectories (e.g., the PNSR curve in Fig. 1) is out of the question. We consider tracking NR-IQMs as a family of baseline methods in Sec. 3.1; the performance is much worse than ours; **(3) Tuning the iteration number**: This ad-hoc solution is taken by most previous works. But since the peak iterations of DIP vary considerably across images and tasks (see, e.g., Figs. 3 and 23 and Appendices A.7.3 and A.7.5), this might entail numerous trial-and-error steps and lead to suboptimal stopping points; **(4) Validation-based ES**: ES easily reminds us of validation-based ES in supervised learning. The DIP approach to IPs as summarized in Eq. (2) **is not** supervised learning, as it only deals with a single instance $y$, without separate $(x, y)$ pairs as training data. There are recent ideas (Yaman et al., 2021; Ding et al., 2022) that hold part of the observation $y$ out as a validation set to emulate validation-based ES in supervised learning, but they quickly become problematic for nonlinear IPs due to the significant violation of the underlying iid assumption; see Sec. 3.3.

**Prior work addressing the overfitting**    There are three main approaches to countering overfitting in working with DIP models. **(1) Regularization**: Heckel & Hand (2019) mitigates overfitting by restricting the size of $G_\theta$ to the underparameterized regime. Metzler et al. (2018); Shi et al. (2022); Jo et al. (2021); Cheng et al. (2019) control the network capacity by regularizing the norms of layerwise weights or the network Jacobian. Liu et al. (2019); Mataev et al. (2019); Sun (2020); Cascarano et al. (2021) use additional regularizer(s) $R(G_\theta(z))$, such as the total-variation norm or trained denoisers. However, in general, it is difficult to choose the right regularization-level to preserve the peak performance while avoiding overfitting, and the optimal $\lambda$ likely depends on the noise type and level, as shown in Sec. 3.1—the default $\lambda$'s for selected methods in this category

still lead to overfitting for high-level noise. **(2) Noise modeling**: You et al. (2020) models sparse additive noise as an explicit term in their optimization objective. Jo et al. (2021) designs regularizers and ES criteria specific to Gaussian and shot noise. Ding et al. (2021) explores subgradient methods with diminishing step-size schedules for impulse noise with the $\ell_1$ loss, with preliminary success. These methods do not work beyond the noise types and levels they target, whereas our knowledge about the noise in a given visual IP is typically limited. **(3) Early stopping (ES)**: Shi et al. (2022) tracks the progress based on a ratio of no-reference blurriness and sharpness, but the criterion only works for their modified DIP models, as acknowledged by the authors. Jo et al. (2021) provides noise-specific regularizer and ES criterion, but it is unclear how to extend the methods to unknown noise types and levels. Li et al. (2021) proposes monitoring the DIP reconstruction by training a coupled autoencoder. Although its performance is similar to ours, the extra autoencoder training slows down the whole process dramatically; see Sec. 3. Yaman et al. (2021); Ding et al. (2022) emulate validation-based ES in supervised learning by splitting elements of $\boldsymbol{y}$ into training and validation sets so that validation-based ES can be performed. But in IPs, especially nonlinear ones (e.g., in blind image deblurring—BID, $\boldsymbol{y} \approx \boldsymbol{k} * \boldsymbol{x}$ where $*$ is linear convolution), elements of $\boldsymbol{y}$ can be far from being iid and so validation may not work well. Moreover, holding-out part of the observation in $\boldsymbol{y}$ can substantially reduce the peak performance; see Sec. 3.3.

**Our contribution** We advocate the ES approach—**the iteration process stops once a good ES point is detected**, as (1) the regularization and noise modeling approaches, even if effective, often do not improve the peak performance but push it until the last iterations; there could be $\geq 10\times$ more iterations spent than that of climbing to the peak in the original DIP models; (2) both need deep knowledge about the noise type/level, which is practically unknown for most applications. If their key models and hyperparameters are not set appropriately, overfitting probably remains. Then ES is still needed. **In this paper, we build a novel ES criterion for various DIP models simply by tracking the trend of the running variance of the reconstruction sequence**. Our ES method is **(1) Effective**: The gap between our detected and the peak performance, i.e., detection gap, is typically very small, as measured by standard visual quality metrics (PSNR and SSIM); **(2) Efficient**: Per-iteration overhead is a fraction of—the standard version in Algorithm 1, or negligible—the variant in Algorithm 2, relative to the per-iteration cost of Eq. (2); **(3) General**: Our method works well for DIP and its variants, including deep decoder (Heckel & Hand, 2019, DD) and sinusoidal representation networks (Sitzmann et al., 2020, SIREN), on different noisy types/levels and across 5 visual IPs, spanning both linear and nonlinear. Also, our method can be wrapped around several regularization methods, e.g., Gaussian process-DIP (Cheng et al., 2019, GP-DIP), DIP with total variation regularization (Liu et al., 2019; Cascarano et al., 2021, DIP-TV) to perform reasonable ES when they fail to prevent overfitting; **(4) Robust**: Our method is relatively insensitive to the two hyperparameters, i.e., window size and patience number (see Secs. 2, 3 and 3.4 and Appendix A.7.13). By contrast, the hyperparameters of most methods reviewed above are sensitive to the noise type/level.

## 2 OUR EARLY-STOPPING METHOD

**Intuition for our method** We assume: $\boldsymbol{x}$ is the unknown groundtruth visual object of size $N$, $\{\boldsymbol{\theta}^t\}_{t\geq 1}$ is the iterate sequence, and $\{\boldsymbol{x}^t\}_{t\geq 1}$ the reconstruction sequence where $\boldsymbol{x}^t \doteq G_{\boldsymbol{\theta}^t}(\boldsymbol{z})$. Since we do not know $\boldsymbol{x}$, we cannot access the PNSR or any FR-IQM curve. But we observe that (Fig. 2) generally the MSE (resp. PSNR; recall $\mathrm{PSNR}(\boldsymbol{x}^t) = 10\log_{10}\|\boldsymbol{x}\|_\infty^2/\mathrm{MSE}(\boldsymbol{x}^t)$) curve follows a U (resp. bell) shape: $\|\boldsymbol{x}^t - \boldsymbol{x}\|_F^2$ initially drops quickly to a low level, and then climbs back due to the noise effect, i.e., the ELTO phenomenon

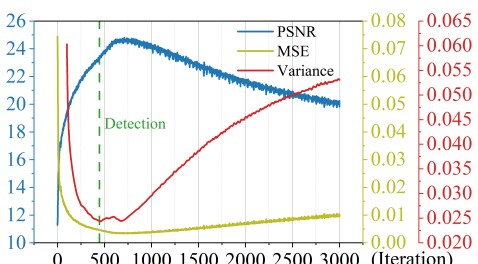

Figure 2: Relationship between the PSNR, MSE, and VAR curves. Our method relies on the VAR curve, whose valley is often well aligned with the MSE valley, to detect the MSE valley—that corresponds to the PSNR peak.

in Sec. 1; we hope to detect the valley of this U-shaped MSE curve. Then how to gauge the MSE curve **without knowing $\boldsymbol{x}$**? We consider the running variance (VAR):

$$\mathrm{VAR}(t) \doteq 1/W \cdot \sum\nolimits_{w=0}^{W-1} \|\boldsymbol{x}^{t+w} - 1/W \cdot \sum\nolimits_{i=0}^{W-1} \boldsymbol{x}^{t+i}\|_F^2. \tag{3}$$

Initially, the models quickly learn the desired visual content, resulting in a monotonic, rapidly decreasing MSE curve (see Fig. 2). So we expect the running variance of $\{\boldsymbol{x}^t\}_{t \geq 1}$ to also drop quickly, as shown in Fig. 2. When the iteration is near the MSE valley, all the $\boldsymbol{x}^t$'s are near but scattered around $\boldsymbol{x}$. So $\frac{1}{W} \sum_{i=0}^{W-1} \boldsymbol{x}^{t+i} \approx \boldsymbol{x}$ and $\mathrm{VAR}(t) \approx \frac{1}{W} \sum_{w=0}^{W-1} \|\boldsymbol{x}^{t+w} - \boldsymbol{x}\|_F^2$. Afterward, the noise effect kicks in and the MSE curve bounces back, leading to a similar bounce-back in the VAR curve as the $\boldsymbol{x}^t$ sequence gradually moves away from $\boldsymbol{x}$.

This argument suggests a U-shaped VAR curve, and the curve should follow the trend of the MSE curve, with approximately aligned valleys, which in turn is aligned with the PSNR peak. To quickly verify this, we randomly sample 1024 images from the RGB track of the NTIRE 2020 Real Image Denoising Chal-

Table 1: ES-WMV (our method) on real-world image denoising for **1024 images**: mean and (std) over the images. (**D**: detected)

| $\ell$ (loss) | PSNR (**D**) | PSNR Gap | SSIM (**D**) | SSIM Gap |
|---|---|---|---|---|
| MSE | 34.04 (3.68) | 0.92 (0.83) | 0.92 (0.07) | 0.02 (0.04) |
| $\ell_1$ | 33.92 (4.34) | 0.92 (0.59) | 0.93 (0.05) | 0.02 (0.02) |
| Huber | 33.72 (3.86) | 0.95 (0.73) | 0.92 (0.06) | 0.02 (0.03) |

lenge (Abdelhamed et al., 2020), and perform DIP-based image denoising (i.e., $\min \ell(\boldsymbol{y}, G_{\boldsymbol{\theta}}(\boldsymbol{z}))$ where $\boldsymbol{y}$ denotes the noisy image). Tab. 1 reports the detected PSNR/SSIM and detection gaps based on our ES method (see Algorithm 1) that tries to detect the valley of the VAR curve. On average, the detection gaps are $\leq 0.95$ in PSNR and $\leq 0.02$ in SSIM, barely noticeable by eyes! More details are in Fig. 11, and Sec. 3 and Appendix A.7.3.

**Detecting transition by running variance**
Our lightweight method only involves computing the VAR curve and numerically detecting its valley—**the iteration stops once the valley is detected**. To obtain the curve, we set a window-size parameter $W$ and compute the windowed moving variance (WMV). To robustly detect the valley, we introduce a patience number $P$ to tolerate up to $P$ consecutive steps of variance stagnation. Obviously, the cost is dominated by the variance calculation per step, which is $O(WN)$ ($N$ is the size of the visual object). In comparison, a typical gradient update step for solving Eq. (2) costs at least $\Omega(|\boldsymbol{\theta}|N)$, where $|\boldsymbol{\theta}|$ is the number of parameters in the DNN $G_{\boldsymbol{\theta}}$. Since $|\boldsymbol{\theta}|$ is typically much larger than $W$ (default: 100), our running VAR and detection incur very little computational overhead. Our whole algorithmic pipeline is summarized in Algorithm 1. To confirm the effectiveness, we provide sample qualitative results in Figs. 3 and 11, with more quantitative results included in the experiment part (Sec. 3;

---

**Algorithm 1** DIP with ES–WMV

**Input:** random seed $\boldsymbol{z}$, randomly-initialized $\boldsymbol{\theta}^0$, window size $W$, patience $P$, empty queue $\mathcal{Q}$, iteration counter $k = 0$, $\mathrm{VAR}_{\min} = \infty$
**Output:** reconstruction $\boldsymbol{x}^*$
1: **while** not stopped **do**
2:     update $\boldsymbol{\theta}$ via Eq. (2) to obtain $\boldsymbol{\theta}^{k+1}$ and $\boldsymbol{x}^{k+1}$
3:     push $\boldsymbol{x}^{k+1}$ to $\mathcal{Q}$, pop queue if $|\mathcal{Q}| > W$
4:     **if** $|\mathcal{Q}| = W$ **then**
5:         compute $\mathrm{VAR}$ of elements in $\mathcal{Q}$ via Eq. (3)
6:         **if** $\mathrm{VAR} < \mathrm{VAR}_{\min}$ **then**
7:             $\mathrm{VAR}_{\min} \leftarrow \mathrm{VAR}$, $\boldsymbol{x}^* \leftarrow \boldsymbol{x}^{k+1}$
8:         **end if**
9:         **if** $\mathrm{VAR}_{\min}$ stagnates for $P$ iterations **then**
10:             stop and return $\boldsymbol{x}^*$
11:         **end if**
12:     **end if**
13:     $k = k + 1$
14: **end while**

---

see also Tab. 1). Appendix A.7.3 shows on image denoising with different noise types/levels, our ES method can detect near-peak ES points. Similarly, our method remains effective on several popular DIP variants, as shown in Fig. 3.

**Seemingly similar ideas** Our running variance and its U-shaped curve are reminiscent of the classical U-shaped bias-variance tradeoff curve and hence validation-based ES (Geman et al., 1992; Yang et al., 2020). But there are crucial differences: (1) our learning setting is not supervised; (2) the variance in supervised learning is with respect to sample distribution, whereas our variance here pertains to the $\{\boldsymbol{x}^t\}_{t \geq 1}$ sequence. As discussed in Sec. 1, we cannot directly apply validation-based ES, although it is possible to heuristically emulate it by splitting the elements in $\boldsymbol{y}$ (Yaman et al., 2021; Ding et al., 2022)—which might be problematic for nonlinear IPs. Another line of related ideas is variance-based online change-point detection in time series analysis (Aminikhanghahi & Cook, 2017), where running variance is often used to detect mean-shift assuming the means are piecewise constant. Here, the piecewise constancy assumption does not hold for our $\{\boldsymbol{x}^t\}_{t \geq 1}$.

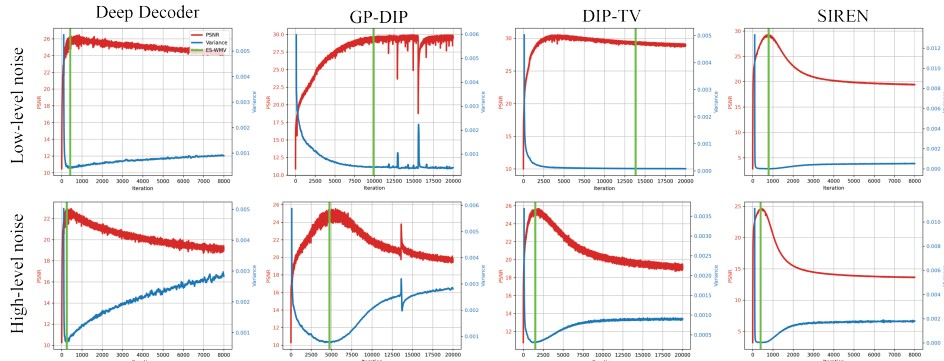

Figure 3: ES-WMV on DD, GP-DIP, DIP-TV, and SIREN for denoising "F16" with different levels of Gaussian noise (top: low-level noise; bottom: high-level noise). Red curves are PSNR curves, and blue curves are VAR curves. The green bars indicate the detected ES points. (We sketch the details of the DIP variants above in Appendix A.5)

**Partial theoretical justification**   We can make our heuristic argument in Sec. 2 more rigorous by restricting ourselves to additive denoising, i.e., $\boldsymbol{y} = \boldsymbol{x} + \boldsymbol{n}$, and appealing to the popular linearization strategy (i.e., neural tangent kernel Jacot et al. (2018); Heckel & Soltanolkotabi (2020b)) in understanding DNNs. The idea is based on the assumption that during DNN training $\boldsymbol{\theta}$ does not move much away from initialization $\boldsymbol{\theta}^0$, so that the learning dynamic can be approximated by that of a linearized model, i.e., suppose that we take the MSE loss

$$\left\|\boldsymbol{y} - G_{\boldsymbol{\theta}}(\boldsymbol{z})\right\|_2^2 \approx \left\|\boldsymbol{y} - G_{\boldsymbol{\theta}^0}(\boldsymbol{z}) - \boldsymbol{J}_G(\boldsymbol{\theta}^0)(\boldsymbol{\theta} - \boldsymbol{\theta}^0)\right\|_2^2 \doteq \widehat{f}(\boldsymbol{\theta}), \tag{4}$$

where $\boldsymbol{J}_G(\boldsymbol{\theta}^0)$ is the Jacobian of $G$ with respect to $\boldsymbol{\theta}$ at $\boldsymbol{\theta}^0$, and $G_{\boldsymbol{\theta}^0}(\boldsymbol{z}) + \boldsymbol{J}_G(\boldsymbol{\theta}^0)(\boldsymbol{\theta} - \boldsymbol{\theta}^0)$ is the first-order Taylor approximation to $G_{\boldsymbol{\theta}}(\boldsymbol{z})$ around $\boldsymbol{\theta}^0$. $\widehat{f}(\boldsymbol{\theta})$ is simply a least-squares objective. We can directly calculate the running variance based on the linear model, as shown below.

**Theorem 2.1.** *Let $\sigma_i$'s and $\boldsymbol{w}_i$'s be the singular values and left singular vectors of $\boldsymbol{J}_G(\boldsymbol{\theta}^0)$, and suppose we run gradient descent with step size $\eta$ on the linearized objective $\widehat{f}(\boldsymbol{\theta})$ to obtain $\{\boldsymbol{\theta}^t\}$ and $\{\boldsymbol{x}^t\}$ with $\boldsymbol{x}^t \doteq G_{\boldsymbol{\theta}^0}(\boldsymbol{z}) + \boldsymbol{J}_G(\boldsymbol{\theta}^0)(\boldsymbol{\theta}^t - \boldsymbol{\theta}^0)$. Then provided that $\eta \leq 1/\max_i(\sigma_i^2)$,*

$$\mathrm{VAR}(t) = \sum_i C_{W,\eta,\sigma_i} \left\langle \boldsymbol{w}_i, \widehat{\boldsymbol{y}} \right\rangle^2 \left(1 - \eta\sigma_i^2\right)^{2t}, \tag{5}$$

*where $\widehat{\boldsymbol{y}} = \boldsymbol{y} - G_{\boldsymbol{\theta}^0}(\boldsymbol{z})$, and $C_{W,\eta,\sigma_i} \geq 0$ only depends on $W$, $\eta$, and $\sigma_i$ for all $i$.*

The proof can be found in Appendix A.2. Theorem 2.1 shows that if the learning rate (LR) $\eta$ is sufficiently small, the WMV of $\{\boldsymbol{x}^t\}$ is monotonically decreasing. We can develop a complementary upper bound for the WMV that does have a U shape. To this end, we make use of Theorem 1 of Heckel & Soltanolkotabi (2020b), which can be summarized (some technical details omitted; precise statement reproduced in Appendix A.3) as follows: consider the two-layer model $G_C(\boldsymbol{B}) = \mathrm{ReLU}(\boldsymbol{UBC})\boldsymbol{v}$, where $\boldsymbol{C} \in \mathbb{R}^{n \times k}$ models $1 \times 1$ trainable convolutions, $\boldsymbol{v} \in \mathbb{R}^{k \times 1}$ contains fixed weights, $\boldsymbol{U}$ is an upsampling operation, and $\boldsymbol{B}$ is the fixed random seed. Let $\boldsymbol{J}$ be a reference Jacobian matrix solely determined by the upsampling operation $\boldsymbol{U}$, and $\sigma_i$'s and $\boldsymbol{w}_i$'s the singular values and left singular vectors of $\boldsymbol{J}$. Assume $\boldsymbol{x} \in \mathrm{span}\{\boldsymbol{w}_1, \ldots, \boldsymbol{w}_p\}$. Then, when $\eta$ is sufficiently small, with high probability,

$$\left\|G_{C^t}(\boldsymbol{B}) - \boldsymbol{x}\right\|_2 \leq \left(1 - \eta\sigma_p^2\right)^t \|\boldsymbol{x}\|_2 + E(\boldsymbol{n}) + \varepsilon\|\boldsymbol{y}\|_2, \tag{6}$$

where $\varepsilon > 0$ is a small scalar related to the structure of the network and $E(\boldsymbol{n})$ is the error introduced by noise: $E^2(\boldsymbol{n}) \doteq \sum_{j=1}^n ((1 - \eta\sigma_j^2)^t - 1)^2 \langle \boldsymbol{w}_j, \boldsymbol{n} \rangle^2$. So if the gap $\sigma_p/\sigma_{p+1} > 1$, $\left\|G_{C^t}(\boldsymbol{B}) - \boldsymbol{x}\right\|_2$ is dominated by $\left(1 - \eta\sigma_p^2\right)^t \|\boldsymbol{x}\|_2$ when $t$ is small, and then by $E(\boldsymbol{n})$ when $t$ is large. But since the former decreases and the latter increases when $t$ grows, the upper bound has a U shape with respect to $t$. Based on this result, we have:

**Theorem 2.2.** *Assume the same setting as Theorem 2 of Heckel & Soltanolkotabi (2020b). With high probability, our WMV is upper bounded by*

$$\frac{12}{W}\|\boldsymbol{x}\|_2^2 \frac{\left(1 - \eta\sigma_p^2\right)^{2t}}{1 - (1 - \eta\sigma_p^2)^2} + 12\sum_{i=1}^n \left(\left(1 - \eta\sigma_i^2\right)^{t+W-1} - 1\right)^2 (\boldsymbol{w}_i^\mathsf{T}\boldsymbol{n})^2 + 12\varepsilon^2\|\boldsymbol{y}\|_2^2. \tag{7}$$

The exact statement and proof can be found in Appendix A.3. By similar reasoning as above, we can conclude that the upper bound in Theorem 2.2 also has a U shape. To interpret the results, Fig. 4 shows the curves (as functions of $t$) predicted by Theorems 2.1 and 2.2. The actual VAR curve should lie between the two curves. These results are primitive and limited, simiar to the situations for many DL theories that provide untight upper and lower bounds; we leave a complete theoretical justification as future work.

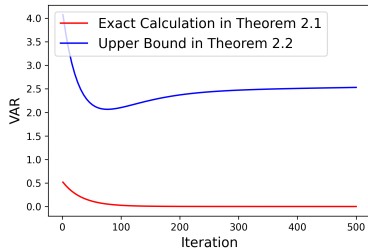

Figure 4: The exact and upper bounds predicted by Theorems 2.1 and 2.2.

**A memory-efficient variant**    While Algorithm 1 is already lightweight and effective in practice, we can slightly modify it to avoid maintaining $\mathcal{Q}$ and hence save memory. The trick is to use exponential moving variance (EMV), together with the exponential moving average (EMA), shown in Appendix A.4. The hard window size parameter $W$ is now replaced by the soft forgetting factor $\alpha$: the larger the $\alpha$, the smaller the impact of the history, and hence a smaller effective window. We compare ES-WMV with ES-EMV in Appendix A.7.11 systematically for image denoising tasks. The latter has slightly better detection due to the strong smoothing effect ($\alpha = 0.1$). For this paper, we prefer to remain simple and leave systematic evaluations of ES-EMV on other IPs as future work.

## 3 EXPERIMENTS

We test ES-WMV for DIP on **image denoising, inpainting, super-resolution, MRI reconstruction, and blind image deblurring**, spanning both linear and nonlinear IPs. For image denoising, we also systematically evaluate ES-WMV on major variants of DIP, including DD (Heckel & Hand, 2019), DIP-TV (Cascarano et al., 2021), GP-DIP (Cheng et al., 2019), and demonstrate ES-WMV as a reliable helper to detect good ES points. Details of the DIP variants are discussed in Appendix A.5. We also compare ES-WMV with major competing methods, including DF-STE (Jo et al., 2021), SV-ES (Li et al., 2021), DOP (You et al., 2020), SB (Shi et al., 2022), and VAL (Yaman et al., 2021; Ding et al., 2022). Details of major ES-

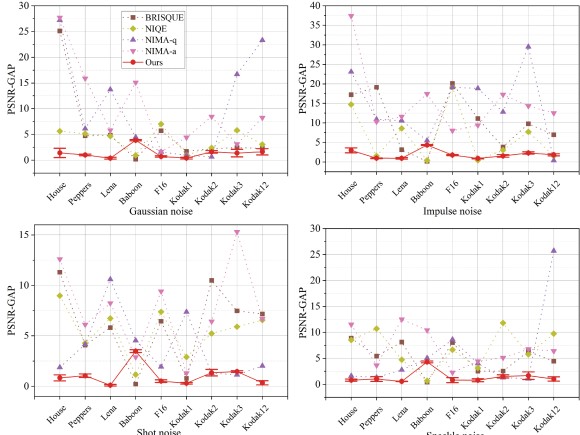

Figure 5: Baseline ES vs our ES-WMV on denoising with **low-level noise**. For NIMA, we report both technical quality assessment (NIMA-q) and aesthetic assessment (NIMA-a). Smaller PSNR gaps are better.

based methods can be found in Appendix A.6. We use both PSNR and SSIM to access the reconstruction quality, and we report PSNR and SSIM gaps (the difference between our detected and peak numbers) as indicators of our detection performance. **Common acronyms, pointers to external codes, detailed experiment settings, results on real-world denoising, inpainting, and super-resolution are in Appendices A.1, A.7.1, A.7.2, A.7.7, A.7.9 and A.7.10, respectively.**

### 3.1 IMAGE DENOISING

Prior works dealing with DIP overfitting mostly focus on image denoising, but typically only evaluate their methods on one or two kinds of noise with low noise levels, e.g., low-level Gaussian noise. To stretch our evaluation, we consider 4 types of noise: Gaussian, shot, impulse, and speckle. We take the classical 9-image dataset (Dabov et al., 2008), and for each noise type, generate two noise levels, low and high, i.e., level 2 and 4 of Hendrycks & Dietterich (2019), respectively. See also the performance of our ES-WMV on real-world denoising in Tab. 1 and Appendix A.7.7.

**Comparison with baseline ES methods**    It is natural to expect that NR-IQMs, such as the classical BRISQUE (Mittal et al., 2012), NIQE (Mittal et al., 2013), and modern DNN-based NIMA (Esfandarani & Milanfar, 2018) can possibly make good ES criteria. We thus set up 3 baseline methods

using BRISQUE, NIQE, and NIMA, respectively and seek the optimal $x^t$ by these metrics. Fig. 5 presents the comparison (in terms of PSNR gaps) of these 3 methods with our ES-WMV on denoising with low-level noise; results on high-level noise, and measured by SSIM are included in Appendix A.7.4. While **our method enjoys favorable detection gaps ($\leq 2$)** for most tested noise types/levels (except for Baboon, Kodak1, Kodak2 for certain noise types/levels; DIP itself is suboptimal in terms of denoising such images with substantial high-frequency components), **detection gaps by the baseline methods can get huge ($\geq 10$).**

**Competing methods**  DF-STE (Jo et al., 2021) is specific for Gaussian and Poisson denoising, and the noise variance is needed for their tuning parameters. Fig. 6 presents the comparison with DF-STE in terms of PSNR. SSIM results are in Appendix A.7.5. Here, we directly report the final PSNRs obtained by both methods. For low-level noise, there is no clear winner. **For high-level noise, ES-WMV**

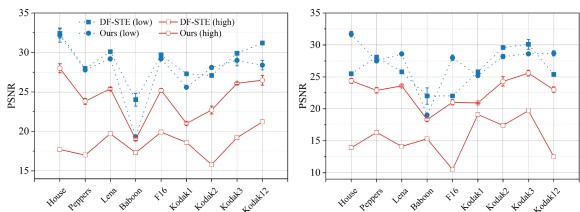

Figure 6: Comparison of DF-STE and ES-WMV for Gaussian and shot noise in terms of PSNR.

**outperforms DF-STE by considerable margins.** Although the right variance level is provided to DF-STE in order to tune their regularization parameters, DF-STE stops after only very few epochs leading to very low performance and almost zero standard deviations—they return almost the noisy input. However, we do not perform any parameter tuning for ES-WMV. We further compare the two methods on CBSD68 in Appendix A.7.5.

We report the results of SV-ES in Appendix A.7.5 since ES-WMV performs largely comparably to SV-ES. However, ES-WMV is much faster in wall-clock time, as reported in Tab. 2: for each epoch, the overhead of our ES-WMV is less than $3/4$ of the DIP update itself, while SV-ES is around $25\times$ of that. There is no

Table 2: Wall-clock time (secs) of DIP and three ES methods per epoch on *NVIDIA Tesla K40 GPU*: mean and (std). Total wall-clock time should contain both DIP and a certain ES method.

|  | DIP | SV-ES | ES-WMV | ES-EMV |
|---|---|---|---|---|
| Time | 0.448 (0.030) | **13.027 (3.872)** | 0.301 (0.016) | 0.003 (0.003) |

surprise: while our method only needs to update the running variance of the $\{x^t\}_{t\geq 1}$ each time, **SV-ES needs to train a coupled autoencoder which is extremely expensive.**

DOP is **designed specifically just for impulse noise**, so we compare ES-WMV with DOP on impulse noise (see Appendix A.7.5). The loss is changed to $\ell_1$ to account for the sparse noise. In terms of the final PSNRs, DOP outperforms DIP with ES-WMV by a small gap, but even the peak PSNR of DIP with $\ell_1$ lags behind DOP by about 2dB for high noise levels.

Table 3: Comparison between ES-WMV and SB for image denoising on the CBSD68 dataset with varying noise level $\sigma$. Higher detected PSNR and earlier detection are better, which are in red: mean and (std).

|  | $\sigma = 15$ | | $\sigma = 25$ | | $\sigma = 50$ | |
|---|---|---|---|---|---|---|
|  | PSNR | Epoch | PSNR | Epoch | PSNR | Epoch |
| WMV | 28.7(3.2) | 3962(2506) | 27.4(2.6) | 3068(2150) | 24.2(2.3) | 1548(1939) |
| SB | 29.0(3.1) | 4908(1757) | 27.3(2.2) | 5099(1776) | 23.0(1.0) | 5765(1346) |

**The ES method in SB is acknowledged to fail for vanilla DIP.** Moreover, their modified model still suffers from the overfitting issue beyond the very low noise levels, as shown in Fig. 20. Their ES method fails to stop at appropriate places when the noise level is high. Hence, **we test both ES-WMV and SB on their modified DIP model** in (Shi et al., 2022), based on two datasets they test: the classic 9-image dataset (Dabov et al., 2008) and CBSD68 dataset (Martin et al., 2001). Qualitative results on the 9 images are shown in Appendix A.7.5; detected PSNR and stopping epochs on the CBSD68 dataset are reported in Tab. 3. For SB, the detection threshold parameter is fixed at $0.01$. It is evident that both methods have similar detection performance for low noise levels but ES-WMV outperforms SB when the noise level is high. Also, ES-WMV tends to stop much earlier than SB, saving computational cost.

We compare VAL with our ES-WMV on the 9-image dataset with low/high-level Gaussian and impulse noise. Since Ding et al. (2022) takes $90\%$ pixels to train DIP and that usually decreases the peak performance, we report the final PSNRs detected by both methods (See Fig. 7). The two ES methods **perform very comparably in image denoising**, which is probably due to the mild violation of the iid assumption only, and also relatively low-degree information loss due to data splitting. **The more complex nonlinear BID in Sec. 3.3 reveals their gap.**

**ES-WMV as a helper for DIP variants**
DD, DIP+TV, GP-DIP represent different regularization strategies for controlling overfitting. A critical issue, however, is setting the right hyperparameters for them so that overfitting is removed while peak-level performance is preserved. So practically, these methods are not free from overfitting, especially when the noise level

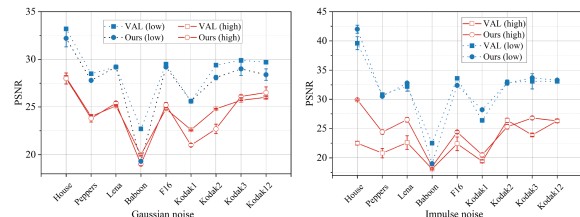

Figure 7: Comparison of VAL and ES-WMV for Gaussian and impulse noise in terms of PSNR.

is high. Thus, instead of treating them as competitors, we test if ES-WMV can reliably detect good ES points for them. We focus on Gaussian denoising, and report the results in Fig. 8 (a)-(c) and Appendix A.7.6. **ES-WMV is able to attain $\leq 1$ PNSR gap for most of the cases**, with few outliers. These regularizations typically change the recovery trajectory. We suspect that finetuning of our method may improve on these corner cases.

**ES-WMV as a helper for implicit neural representations (INRs)** INRs, such as Tancik et al. (2020) and Sitzmann et al. (2020), use multilayer perceptrons to represent high-frequency functions in low-dimensional problem domains and have achieved superior results on complex 3D visual tasks. We further extend our ES-WMV to help the INR family and take SIREN (Sitzmann et al., 2020) as an example. SIREN parameterizes $x$ as the discretization of a continuous function: this function takes into spatial coordinates and returns the corresponding function values. Here, we test SIREN, which is reviewed in Appendix A.5, as a replacement of DIP models for Gaussian denoising, and summarize the results in Fig. 8 and Fig. 21. **ES-WMV is again able to detect near-peak performance for most images.**

(a) Deep Decoder  (b) GP-DIP

(c) DIP-TV  (d) SIREN

Figure 8: Performance of ES-WMV on DD, GP-DIP, DIP-TV, and SIREN for Gaussian denoising in terms of PSNR gaps. L: low noise level; H: high noise level.

### 3.2 MRI RECONSTRUCTION

We further test ES-WMV on MRI reconstruction, a classical linear IP with a nontrivial forward mapping: $y \approx \mathcal{F}(x)$, where $\mathcal{F}$ is the subsampled Fourier

Table 4: ConvDecoder on MRI reconstruction for **30 cases**: mean and (std). (**D**: Detected)

| PSNR(**D**) | PSNR Gap | SSIM(**D**) | SSIM Gap |
|---|---|---|---|
| 32.63 (2.36) | 0.23 (0.32) | 0.81 (0.09) | 0.01 (0.01) |

operator, and we use $\approx$ to indicate that the noise encountered in practical MRI imaging may be hybrid (e.g., additive, shot) and uncertain. Here, we take 8-fold undersampling and parametrize $x$ using "Conv-Decoder" (Darestani & Heckel, 2021), a variant of DD. Due to the heavy overparameterization, overfitting occurs, and ES is needed. Darestani & Heckel (2021) directly sets the stopping point at the 2500-th epoch, and we run our ES-WMV. We visualize the performance on two random cases (C1: 1001339 and C2: 1000190 sampled from Darestani & Heckel (2021), part of the fastMRI datatset (Zbontar et al., 2018)) in Fig. 23 (quality measured in SSIM, consistent with Darestani & Heckel (2021)). It is clear that ES-WMV detects near-peak performance for both cases, and it is adaptive enough to yield comparable or better ES points than heuristically fixed ES points. We further test our ES-WMV on ConvDecoder for **30 cases** from the fastMRI dataset (see Tab. 4), which shows the precise and stable detection of ES-WMV.

### 3.3 BLIND IMAGE DEBLURRING (BID)

In BID, a blurry and noisy image is given, and the goal is to recover a sharp and clean image. The blur is mostly caused by motion and/or optical nonideality in the camera, and the forward process

is often modeled as $\boldsymbol{y} = \boldsymbol{k} * \boldsymbol{x} + \boldsymbol{n}$, where $\boldsymbol{k}$ is the blur kernel, $\boldsymbol{n}$ models additive sensory noise, and $*$ is linear convolution to model the spatial uniformity of the blur effect (Szeliski, 2022). BID is a very challenging visual IP due to the bilinearity: $(\boldsymbol{k}, \boldsymbol{x}) \mapsto \boldsymbol{k} * \boldsymbol{x}$. Recently, Ren et al. (2020); Wang et al. (2019); Asim et al. (2020); Tran et al. (2021) have tried to use DIP models to solve BID by modeling $\boldsymbol{k}$ and $\boldsymbol{x}$ as two separate DNNs, i.e., $\min_{\boldsymbol{\theta}_k, \boldsymbol{\theta}_x} \|\boldsymbol{y} - G_{\boldsymbol{\theta}_k}(\boldsymbol{z}_k) * G_{\boldsymbol{\theta}_x}(\boldsymbol{z}_x)\|_2^2 + \lambda \|\nabla G_{\boldsymbol{\theta}_x}(\boldsymbol{z}_x)\|_1 / \|\nabla G_{\boldsymbol{\theta}_x}(\boldsymbol{z}_x)\|_2$, where the regularizer is to promote sparsity in the gradient domain for reconstruction of $\boldsymbol{x}$, as standard in BID. We follow Ren et al. (2020) and choose multi-layer perceptron (MLP) with softmax activation for $G_{\boldsymbol{\theta}_k}$, and the canonical DIP model (CNN-based encoder-decoder architecture) for $G_{\boldsymbol{\theta}_x}(\boldsymbol{z}_x)$. We change their regularizer from the original $\|\nabla G_{\boldsymbol{\theta}_x}(\boldsymbol{z}_x)\|_1$ to the current, as their original formulation is tested only on a very low noise level $\sigma = 10^{-5}$ and no overfitting is observed. We set to work with higher noise level $\sigma = 10^{-3}$, and find that their original formulation does not work. The positive effect of the modified regularizer on BID is discussed in Krishnan et al. (2011).

First, we take 4 images and 3 kernels from the standard Levin dataset (Levin et al., 2011), resulting in 12 image-kernel combinations. The high noise level leads to substantial overfitting, as shown in Fig. 9 (top left). Nonetheless, ES-WMV can reliably detect good ES points and lead to impressive visual reconstructions (see Fig. 9 (top right)). We systematically compare VAL and our ES-WMV on this difficult nonlinear IP, as we suspect that nonlinearity can break VAL down as discussed in Sec. 1, and subsampling the observation $\boldsymbol{y}$ for training-validation splitting may be unwise. Our results (Fig. 9 (bottom left/right)) confirm these predictions: the peak performance is much worse after $10\%$ of elements in $\boldsymbol{y}$ are removed for valiation. In contrast, our ES-WMV returns quantitatively

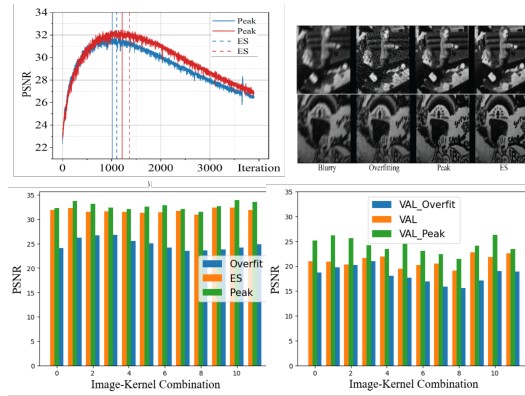

Figure 9: Top left: ES-WMV on BID; Top right: visual results of ES-WMV; Bottom: quantitative results of ES-WMV and VAL, respectively

near-peak performance, far better than leaving the process to overfit. In Appendix A.7.12, we test both low- and high-level noise on the entire Levin dataset for completeness.

## 3.4 ABLATION STUDY

The window size $W$ (default: 100) and patience number $P$ (default: 1000) are the only hyperparameters for ES-WMV. To study their impact on ES detection, we vary them across a range and check how the detection gap changes for Gaussian denoising on the classic 9-image

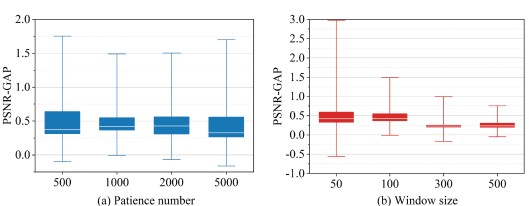

Figure 10: Effect of $W$ and $P$

dataset (Dabov et al., 2008) with medium-level noise, as shown in Fig. 10 for PSNR gaps and Fig. 26 for SSIM gaps. Our method is robust against these changes, and it seems larger $W$ and $P$ can bring in marginal improvement.

## 4 DISCUSSION

We have proposed a simple yet effective ES detection method (ES-WMV, and the ES-EMV variant) that works robustly across multiple visual IPs and DIP variants. In comparison, competing ES methods are noise- or DIP-model-specific, and only work for limited scenarios; Li et al. (2021) has comparable performance but it slows down the running speed too much; validation-based ES (Ding et al., 2022) works well for the simple denoising task while lags behind our ES method a lot in nonlinear IPs, e.g., BID. As for limitations, our theoretical justification is only partial, sharing the same difficulty of analyzing DNNs in general. Our ES method struggles with images with substantial high-frequency components; DIP needs to run numerous iterative steps for every instance, which is not ideal for time-constrained applications.

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

## A  APPENDIX

### A.1  ACRONYMS

| **List of Common Acronyms** (in alphabetic order) | |
| --- | --- |
| CI | computational imaging |
| CNN | convolutional neural network |
| DD | deep decoder |
| DIP | deep image prior |
| DIP-TV | DIP with total variation regularization |
| DL | deep learning |
| DNN | deep neural network |
| ELTO | early-learning-then-overfitting |
| ES | early stopping |
| EMA | exponential moving average |
| EMV | exponential moving variance |
| FR-IQM | full-reference image quality metric |
| GP-DIP | Gaussian process DIP |
| INR | implicit neural representations |
| IP | inverse problem |
| MSE | mean squared error |
| NR-IQM | no-reference image quality metric |
| PSNR | peak signal-to-noise ratio |
| SIREN | sinusoidal representation networks |
| SOTA | state-of-the-art |
| VAR | variance |
| WMV | windowed moving variance |

### A.2  PROOF OF 2.1

*Proof.* To simplify the notation, we write $\widehat{\boldsymbol{y}} \doteq \boldsymbol{y} - G_{\boldsymbol{\theta}^0}(\boldsymbol{z})$, $\boldsymbol{J} \doteq \boldsymbol{J}_G(\boldsymbol{\theta}^0)$, and $\boldsymbol{c} \doteq \boldsymbol{\theta} - \boldsymbol{\theta}^0$. So the least-squares objective in Eq. (4) is equivalent to

$$\|\widehat{\boldsymbol{y}} - \boldsymbol{J}\boldsymbol{c}\|_2^2 \tag{8}$$

and the gradient update reads

$$\boldsymbol{c}^t = \boldsymbol{c}^{t-1} - \eta \boldsymbol{J}^\intercal \big( \boldsymbol{J}\boldsymbol{c}^{k-1} - \widehat{\boldsymbol{y}} \big), \tag{9}$$

where $\boldsymbol{c}^0 = \boldsymbol{0}$ and $\boldsymbol{x}^t = \boldsymbol{J}\boldsymbol{c}^t + G_{\boldsymbol{\theta}^0}(\boldsymbol{z})$. The residual at time $t$ can be computed as

$$\boldsymbol{r}^t \doteq \widehat{\boldsymbol{y}} - \boldsymbol{J}\boldsymbol{c}^t \tag{10}$$

$$= \widehat{\boldsymbol{y}} - \boldsymbol{J} \big( \boldsymbol{c}^{t-1} - \eta \boldsymbol{J}^\intercal \big( \boldsymbol{J}\boldsymbol{\theta}^{t-1} - \widehat{\boldsymbol{y}} \big) \big) \tag{11}$$

$$= (\boldsymbol{I} - \eta \boldsymbol{J}\boldsymbol{J}^\intercal) \big( \widehat{\boldsymbol{y}} - \boldsymbol{J}\boldsymbol{c}^{t-1} \big) \tag{12}$$

$$= (\boldsymbol{I} - \eta \boldsymbol{J}\boldsymbol{J}^\mathsf{T})^2 \left(\widehat{\boldsymbol{y}} - \boldsymbol{J}\boldsymbol{c}^{t-2}\right) = \dots \tag{13}$$

$$= (\boldsymbol{I} - \eta \boldsymbol{J}\boldsymbol{J}^\mathsf{T})^t \left(\widehat{\boldsymbol{y}} - \boldsymbol{J}\boldsymbol{c}^0\right) \quad (\text{using } \boldsymbol{c}^0 = \boldsymbol{0}) \tag{14}$$

$$= (\boldsymbol{I} - \eta \boldsymbol{J}\boldsymbol{J}^\mathsf{T})^t \widehat{\boldsymbol{y}}. \tag{15}$$

Assume the SVD of $\boldsymbol{J}$ as $\boldsymbol{J} = \boldsymbol{W}\boldsymbol{\Sigma}\boldsymbol{V}^\mathsf{T}$. Then

$$\boldsymbol{r}^t = \left(\boldsymbol{I} - \eta \boldsymbol{W}\boldsymbol{\Sigma}^2\boldsymbol{W}^\mathsf{T}\right)^t \widehat{\boldsymbol{y}} = \sum_i \left(1 - \eta\sigma_i^2\right)^t \boldsymbol{w}_i^\mathsf{T}\widehat{\boldsymbol{y}}\boldsymbol{w}_i \tag{16}$$

and so

$$\boldsymbol{J}\boldsymbol{c}^t = \widehat{\boldsymbol{y}} - \boldsymbol{r}^t = \sum_i \left(1 - \left(1 - \eta\sigma_i^2\right)^t\right) \boldsymbol{w}_i^\mathsf{T}\widehat{\boldsymbol{y}}\boldsymbol{w}_i. \tag{17}$$

Consider a set of $W$ vectors $\mathcal{V} = \{\boldsymbol{v}_1, \dots, \boldsymbol{v}_W\}$. We have that the empirical variance

$$\text{VAR}(\mathcal{V}) = \frac{1}{W}\sum_{w=1}^{W} \left\| \boldsymbol{v}_w - \frac{1}{W}\sum_{j=1}^{W} \boldsymbol{v}_j \right\|_2^2 = \frac{1}{W}\sum_{w=1}^{W} \|\boldsymbol{v}_w\|_2^2 - \left\| \frac{1}{W}\sum_{w=1}^{W} \boldsymbol{v}_w \right\|_2^2. \tag{18}$$

So the variance of the set $\{\boldsymbol{x}^t, \boldsymbol{x}^{t+1}, \dots, \boldsymbol{x}^{t+W-1}\}$, same as the variance of the set $\{\boldsymbol{J}\boldsymbol{c}^t, \boldsymbol{J}\boldsymbol{c}^{t+1}, \dots, \boldsymbol{J}\boldsymbol{c}^{t+W-1}\}$, can be calculated as

$$\frac{1}{W}\sum_{w=0}^{W-1}\sum_i (\boldsymbol{w}_i^\mathsf{T}\widehat{\boldsymbol{y}})^2 \left(1 - \left(1 - \eta\sigma_i^2\right)^{t+w}\right)^2 - \frac{1}{W^2}\sum_i (\boldsymbol{w}_i^\mathsf{T}\widehat{\boldsymbol{y}})^2 \left(\sum_{w=0}^{W-1} 1 - \left(1 - \eta\sigma_i^2\right)^{t+w}\right)^2 \tag{19}$$

$$= \frac{1}{W^2}\sum_i (\boldsymbol{w}_i^\mathsf{T}\widehat{\boldsymbol{y}})^2 \left[ W\sum_{w=0}^{W-1} \left(1 - \left(1 - \eta\sigma_i^2\right)^{t+w}\right)^2 - \left(\sum_{w=0}^{W-1} 1 - \left(1 - \eta\sigma_i^2\right)^{t+w}\right)^2 \right] \tag{20}$$

$$= \frac{1}{W^2}\sum_i (\boldsymbol{w}_i^\mathsf{T}\widehat{\boldsymbol{y}})^2 \left[ \left(W^2 + W\frac{(1 - \eta\sigma_i^2)^{2t}(1 - (1 - \eta\sigma_i^2)^{2W})}{1 - (1 - \eta\sigma_i^2)^2} - 2W\frac{(1 - \eta\sigma_i^2)^t(1 - (1 - \eta\sigma_i^2)^W)}{\eta\sigma_i^2}\right) \right.$$
$$\left. - \left(W^2 - 2W\frac{(1 - \eta\sigma_i^2)^t(1 - (1 - \eta\sigma_i^2)^W)}{\eta\sigma_i^2} + \frac{(1 - \eta\sigma_i^2)^{2t}\left(1 - (1 - \eta\sigma_i^2)^W\right)^2}{\eta^2\sigma_i^4}\right) \right] \tag{21}$$

$$= \frac{1}{W^2}\sum_i \langle \boldsymbol{w}_i, \widehat{\boldsymbol{y}}\rangle^2 \frac{(1 - \eta\sigma_i^2)^{2t}}{\eta\sigma_i^2} \left[ W\frac{1 - (1 - \eta\sigma_i^2)^{2W}}{2 - \eta\sigma_i^2} - \frac{(1 - (1 - \eta\sigma_i^2)^W)^2}{\eta\sigma_i^2} \right]. \tag{22}$$

So the constants $C_{W,\eta,\sigma_i}$'s are defined as

$$C_{W,\eta,\sigma_i} \doteq \frac{1}{W^2\eta\sigma_i^2} \left[ W\frac{1 - (1 - \eta\sigma_i^2)^{2W}}{2 - \eta\sigma_i^2} - \frac{(1 - (1 - \eta\sigma_i^2)^W)^2}{\eta\sigma_i^2} \right]. \tag{23}$$

To see they are nonnegative, it is sufficient to show that

$$W\frac{1 - (1 - \eta\sigma_i^2)^{2W}}{2 - \eta\sigma_i^2} - \frac{(1 - (1 - \eta\sigma_i^2)^W)^2}{\eta\sigma_i^2} \geq 0$$

$$\iff \eta\sigma_i^2 W\left(1 - (1 - \eta\sigma_i^2)^{2W}\right) - \left(2 - \eta\sigma_i^2\right)(1 - (1 - \eta\sigma_i^2)^W)^2 \geq 0. \tag{24}$$

Now consider the function

$$h(\xi, W) = \xi W\left(1 - (1 - \xi)^{2W}\right) - (2 - \xi)(1 - (1 - \xi)^W)^2 \quad \xi \in [0, 1], W \geq 1. \tag{25}$$

First, one can easily check that $\partial_W h(\xi, W) \geq 0$ for all $W \geq 1$ and all $\xi \in [0, 1]$, i.e., $h(\xi, W)$ is monotonically increasing with respect to $W$. Thus, in order to prove $C_{W,\eta,\sigma_i} \geq 0$, it suffices to show that $h(\xi, 1) \geq 0$. Now

$$h(\xi, 1) = \xi\left(1 - (1 - \xi)^2\right) - (2 - \xi)\xi^2 = 0, \tag{26}$$

completing the proof. $\qquad\square$

A.3 PROOF OF 2.2

We first restate Theorem 2 in Heckel & Soltanolkotabi (2020b).

**Theorem A.1** (Heckel & Soltanolkotabi (2020b)). *Let $x \in \mathbb{R}^n$ be a signal in the span of the first $p$ trigonometric basis functions, and consider a noisy observation $y = x + n$, where the noise $n \sim \mathcal{N}\left(0, \xi^2/n \cdot I\right)$. To denoise this signal, we fit a two-layer generator network $G_C(B) = \text{ReLU}(UBC)v$, where $v = [1, \ldots, 1, -1, \ldots, -1]/\sqrt{k}$, and $B \sim_{iid} \mathcal{N}(0,1)$, and $U$ is an upsampling operator that implements circular convolution with a given kernel $u$. Denote $\sigma \doteq \|u\|_2 |Fg(u \circledast u/\|u\|_2^2)|^{1/2}$ where $g(t) = (1 - \cos^{-1}(t)/\pi)t$ and $\circledast$ denotes the circular convolution. Fix any $\varepsilon \in (0, \sigma_p/\sigma_1]$, and suppose $k \geq C_u n/\varepsilon^8$, where $C_u > 0$ is a constant only depending on $u$. Consider gradient descent with step size $\eta \leq \|Fu\|_\infty^{-2}$ ($Fu$ is the Fourier transform of $u$) starting from $C_0 \sim_{iid} \mathcal{N}\left(0, \omega^2\right)$, entries, $\omega \propto \frac{\|y\|_2}{\sqrt{n}}$. Then, for all iterates $t$ obeying $t \leq \frac{100}{\eta \sigma_p^2}$, the reconstruction error obeys*

$$\|G_{C^t}(B) - x\|_2 \leq \left(1 - \eta \sigma_p^2\right)^t \|x\|_2 + \sqrt{\sum_{i=1}^n \left((1 - \eta \sigma_i^2)^t - 1\right)^2 (w_i^\mathsf{T} n)^2} + \varepsilon \|y\|_2$$

*with probability at least $1 - \exp\left(-k^2\right) - n^{-2}$.*

Note that since $B \sim_{iid} \mathcal{N}(0,1)$ and hence is full-rank with probability one, the original Theorem 1 & 2 of Heckel & Soltanolkotabi (2020b) rename $BC$ into $C'$ and state the result directly on $C'$, i.e., assume the model is $\text{ReLU}(UC')v$. It is easy to see the original theorems imply the version stated here.

With this, we can obtain our Theorem 2.2, stated in full technical form here:

**Theorem A.2.** *Let $x \in \mathbb{R}^n$ be a signal in the span of the first $p$ trigonometric basis functions, and consider a noisy observation $y = x + n$, where the noise $n \sim \mathcal{N}\left(0, \xi^2/n \cdot I\right)$. To denoise this signal, we fit a two-layer generator network $G_C(B) = \text{ReLU}(UBC)v$, where $v = [1, \ldots, 1, -1, \ldots, -1]/\sqrt{k}$, and $B \sim_{iid} \mathcal{N}(0,1)$, and $U$ is an upsampling operator that implements circular convolution with a given kernel $u$. Denote $\sigma \doteq \|u\|_2 |Fg(u \circledast u/\|u\|_2^2)|^{1/2}$ where $g(t) = (1 - \cos^{-1}(t)/\pi)t$ and $\circledast$ denotes the circular convolution. Fix any $\varepsilon \in (0, \sigma_p/\sigma_1]$, and suppose $k \geq C_u n/\varepsilon^8$, where $C_u > 0$ is a constant only depending on $u$. Consider gradient descent with step size $\eta \leq \|Fu\|_\infty^{-2}$ ($Fu$ is the Fourier transform of $u$) starting from $C_0 \sim_{iid} \mathcal{N}\left(0, \omega^2\right)$, entries, $\omega \propto \frac{\|y\|_2}{\sqrt{n}}$. Then, for all iterates $t$ obeying $t \leq \frac{100}{\eta \sigma_p^2}$, our WMV obeys*

$$\text{WMV} \leq \frac{12}{W} \|x\|_2^2 \frac{\left(1 - \eta \sigma_p^2\right)^{2t}}{1 - (1 - \eta \sigma_p^2)^2} + 12 \sum_{i=1}^n \left(\left(1 - \eta \sigma_i^2\right)^{t+W-1} - 1\right)^2 (w_i^\mathsf{T} n)^2 + 12 \varepsilon^2 \|y\|_2^2 \quad (27)$$

*with probability at least $1 - \exp\left(-k^2\right) - n^{-2}$.*

*Proof.* We make use of the basic inequality: $\|a - b\|_2^2 \leq 2\|a\|_2^2 + 2\|b\|_2^2$ for any two vectors $a, b$ of compatible dimension. We have

$$\frac{1}{W} \sum_{w=0}^{W-1} \left\| G_{C^{t+w}}(B) - \frac{1}{W} \sum_{j=0}^{W-1} G_{C^{t+j}}(B) \right\|_2^2 \quad (28)$$

$$= \frac{1}{W} \sum_{w=0}^{W-1} \left\| G_{C^{t+w}}(B) - x + x - \frac{1}{W} \sum_{j=0}^{W-1} G_{C^{t+j}}(B) \right\|_2^2 \quad (29)$$

$$\leq \left( \frac{2}{W} \sum_{w=0}^{W-1} \| G_{C^{t+w}}(B) - x \|_2^2 \right) + 2 \left\| x - \frac{1}{W} \sum_{j=0}^{W-1} G_{C^{t+j}}(B) \right\|_2^2 \quad (30)$$

$$\leq \frac{2}{W} \sum_{w=0}^{W-1} \| G_{C^{t+w}}(B) - x \|_2^2 + \frac{2}{W} \sum_{j=0}^{W-1} \| G_{C^{t+j}}(B) - x \|_2^2 \quad (31)$$

$$(\boldsymbol{z} \mapsto \|\boldsymbol{z} - \boldsymbol{x}\|_2^2 \text{ convex and Jensen's inequality})$$

$$= \frac{4}{W} \sum_{w=0}^{W-1} \|G_{\boldsymbol{C}^{t+w}}(\boldsymbol{B}) - \boldsymbol{x}\|_2^2. \tag{32}$$

In view of Theorem A.1,

$$\|G_{\boldsymbol{C}^{t+w}}(\boldsymbol{B}) - \boldsymbol{x}\|_2^2 \leq 3 \left(1 - \eta\sigma_p^2\right)^{2t+2w} \|\boldsymbol{x}\|_2^2 + 3 \sum_{i=1}^n \left(\left(1 - \eta\sigma_j^2\right)^{t+w} - 1\right)^2 (\boldsymbol{w}_i^\mathsf{T}\boldsymbol{n})^2 + 3\varepsilon^2 \|\boldsymbol{y}\|_2^2. \tag{33}$$

Thus,

$$\sum_{w=0}^{W-1} \|G_{\boldsymbol{C}^{t+w}}(\boldsymbol{B}) - \boldsymbol{x}\|_2^2$$

$$\leq 3\|\boldsymbol{x}\|_2^2 \sum_{w=0}^{W-1} \left(1 - \eta\sigma_p^2\right)^{2t+2w} + 3 \sum_{w=0}^{W-1} \sum_{i=1}^n \left(\left(1 - \eta\sigma_i^2\right)^{t+w} - 1\right)^2 (\boldsymbol{w}_i^\mathsf{T}\boldsymbol{n})^2 + 3W\varepsilon^2\|\boldsymbol{y}\|_2^2 \tag{34}$$

$$\leq 3\|\boldsymbol{x}\|_2^2 \frac{\left(1 - \eta\sigma_p^2\right)^{2t} \left(1 - (1 - \eta\sigma_p^2)^{2W}\right)}{1 - (1 - \eta\sigma_p^2)^2} + 3W \sum_{i=1}^n \left(\left(1 - \eta\sigma_i^2\right)^{t+W-1} - 1\right)^2 (\boldsymbol{w}_i^\mathsf{T}\boldsymbol{n})^2 + 3W\varepsilon^2\|\boldsymbol{y}\|_2^2 \tag{35}$$

$$\leq 3\|\boldsymbol{x}\|_2^2 \frac{\left(1 - \eta\sigma_p^2\right)^{2t}}{1 - (1 - \eta\sigma_p^2)^2} + 3W \sum_{i=1}^n \left(\left(1 - \eta\sigma_i^2\right)^{t+W-1} - 1\right)^2 (\boldsymbol{w}_i^\mathsf{T}\boldsymbol{n})^2 + 3W\varepsilon^2\|\boldsymbol{y}\|_2^2, \tag{36}$$

completing the proof. $\qquad\square$

### A.4 ES-EMV ALGORITHM

The exponential moving variance version of our method is summarized in Algorithm 2.

---

**Algorithm 2** DIP with ES–EMV

---

**Input:** random seed $\boldsymbol{z}$, randomly-initialized $G_{\boldsymbol{\theta}}$, forgetting factor $\alpha \in (0,1)$, patience number $P$, iteration counter $k = 0$, $\mathrm{EMA}^0 = 0$, $\mathrm{EMV}^0 = 0$, $\mathrm{EMV}_{\min} = \infty$
**Output:** reconstruction $\boldsymbol{x}^*$
 1: **while** not stopped **do**
 2:     update $\boldsymbol{\theta}$ via Eq. (2) to obtain $\boldsymbol{\theta}^{k+1}$ and $\boldsymbol{x}^{k+1}$
 3:     $\mathrm{EMA}^{k+1} = (1-\alpha)\mathrm{EMA}^k + \alpha\boldsymbol{x}^{k+1}$
 4:     $\mathrm{EMV}^{k+1} = (1-\alpha)\mathrm{EMV}^k + \alpha(1-\alpha)\|\boldsymbol{x}^{k+1} - \mathrm{EMA}^k\|_2^2$
 5:     **if** $\mathrm{EMV}^{\mathrm{k}+1} < \mathrm{EMV}_{\min}$ **then**
 6:         $\mathrm{EMV}_{\min} \leftarrow \mathrm{EMV}^{\mathrm{k}+1}$, $\boldsymbol{x}^* \leftarrow \boldsymbol{x}^{k+1}$
 7:     **end if**
 8:     **if** $\mathrm{EMV}_{\min}$ stagnates for $P$ iterations **then**
 9:         stop and return $\boldsymbol{x}^*$
10:     **end if**
11:     $k = k + 1$
12: **end while**

---

### A.5 MORE DETAILS ON MAJOR DIP VARIANTS

**Deep Decoder (DD)**   (Heckel & Hand, 2019) differs from DIP mainly in terms of the network architecture: it is typically an *under-parameterized* network consisting of mainly $1 \times 1$ convolutions, upsampling, ReLU and channel-wise normalization layers, while DIP uses an *over-parameterized*, U-net like convolutional network.

**GP-DIP**  (Cheng et al., 2019) uses the original DIP (Ulyanov et al., 2018) network and formulation, but replaces the stochastic gradient descent (SGD) by stochastic gradient Langevin dynamics (SGLD) in the gradient update step. i.e., for generic gradient step for optimizing Eq. (2) reads:

$$\boldsymbol{\theta}^+ = \boldsymbol{\theta} - t\nabla_{\boldsymbol{\theta}}[\ell(\boldsymbol{y}, f(G_{\boldsymbol{\theta}}(\boldsymbol{z}))) + \lambda R(G_{\boldsymbol{\theta}}(\boldsymbol{z}))] + \eta \tag{37}$$

where $\eta$ is zero-mean Gaussian with an isotropic variance level $t$.

**DIP-TV**  (Cascarano et al., 2021) uses the original DIP (Ulyanov et al., 2018) network, with a Total Variation (TV) regularizer added. Then, the proposed objective is solved with Alternating Direction Method of Multipliers (ADMM) framework.

**SIREN**  (Sitzmann et al., 2020) treats the object directly as a continuous function on $\mathbb{R}^2$ or $\mathbb{R}^3$ (or higher-dimensional spaces depending on the application) and hence parameterizes it as a multi-layer perceptron (MLP): 1) the input to SIREN is the 2D/3D coordinate of each pixel instead of random values, and 2) the network uses a sinusoidal activation function instead of the commonly used ReLU. When substituting the DIP network with SIREN and solve Eq. (2) problems, similar overfitting issue is still observed.

### A.6  More details on major ES methods

Here, we provide more details on major competing methods, all of them ES-based except for You et al. (2020).

**Spectral Bias (SB)**  Shi et al. (2022) operates on DD models, and proposes two modifications to change the spectral bias: (1) controlling the operator norm of the weight $\boldsymbol{w}$ for each convolutional layer by the normalization

$$\boldsymbol{w}' = \frac{\boldsymbol{w}}{\max\left(1, \|\boldsymbol{w}\|_{\mathrm{op}}/\lambda\right)}, \tag{38}$$

ensuring that $\|\boldsymbol{w}'\|_{\mathrm{op}} \leq \lambda$, which in turn controls the Fourier spectrum of the underlying function represented by the layer; (2) performing Gaussian upsampling instead of the typical bilinear upsampling to suppress the smoothness effect of the latter. These two modifications with appropriate parameter setting ($\lambda$, and $\sigma$ in Gaussian filtering) can improve the learning of the high-frequency components by DD, and allow the blurriness-over-sharpness stopping criterion

$$\Delta r(\boldsymbol{x}^t) = \frac{1}{W}\left|\sum_{w=1}^{W} r(\boldsymbol{x}^{t-w}) - \sum_{w=1}^{W} r(\boldsymbol{x}^{t-W-w})\right|, \tag{39}$$

where $r(\boldsymbol{x}') = B(\boldsymbol{x}')/S(\boldsymbol{x}')$, and $B(\cdot)$ and $S(\cdot)$ are the blurriness and sharpness metrics in Crete et al. (2007) and Bahrami & Kot (2014), respectively. In other words, the criterion in Eq. (39) measures the change of average blurriness-over-sharpness ratios over consecutive windows of size $W$, and small changes indicate good ES points. But, as said, this criterion only works for the modified DD models and not other DIP variants, as acknowledged by the authors in Shi et al. (2022) and confirmed in our experiment (see Sec. 3.1).

**DF-STE**  Jo et al. (2021) targets Gaussian denoising with known noise levels (i.e., $\boldsymbol{y} = \boldsymbol{x} + \boldsymbol{n}$, where $n$ is iid Gaussian noise), and considers the objective

$$\min_{\boldsymbol{\theta}} \frac{1}{n^2}\|\boldsymbol{y} - G_{\boldsymbol{\theta}}(\boldsymbol{y})\|_F^2 + \frac{\sigma^2}{n^2}\operatorname{tr}\boldsymbol{J}_{G_{\boldsymbol{\theta}}}(\boldsymbol{y}), \tag{40}$$

where $\operatorname{tr}\boldsymbol{J}_{G_{\boldsymbol{\theta}}}(\boldsymbol{y})$ is the trace of the network Jacobian with respect to the input, i.e., the divergence term in Jo et al. (2021). The divergence term is a proxy for controlling the capacity of the network. The paper then proposes a heuristic zero-crossing stopping criterion that stops the iteration when the loss starts to cross zero into negative values. Although the idea works reasonably well on Gaussian denoising with low and known noise level (the variance level $\sigma^2$ is explicitly needed in the regularization parameter ahead of the divergence term), it starts to break down when the noise level increases even if the right noise level is provided; see Sec. 3.1. Also, although the paper has extended the formulation to handle Poisson noise, it is unclear how to generalize the idea for handling other types of noise, as well as how to move beyond simple additive denoising problems.

**SV-ES**  Li et al. (2021) proposes training an autoencoder online using the reconstruction sequence $\{\boldsymbol{x}^t\}_{t\geq 1}$:

$$\min_{\boldsymbol{w},\boldsymbol{v}} \sum_{t\geq 1} \ell_{\mathrm{AE}}\big(\boldsymbol{x}^t, D_{\boldsymbol{w}} \circ E_{\boldsymbol{v}}(\boldsymbol{x}^t)\big). \tag{41}$$

Any new $\boldsymbol{x}^t$ is passed through the current autoencoder, and the reconstruction error $\ell_{\mathrm{AE}}$ is recorded. We observe that the error curve typically follows a U-shape, and the valley of the curve is approximately aligned with the peak of the PNSR curve. We hence design an ES method by detecting the valley of the error curve. This method works reasonably well across different IPs and different DIP variants. A major drawback is the efficiency: the overhead caused by online training of the autoencoder is order-of-magnitude larger than the cost of DIP update itself, as shown in Tab. 2.

**DOP**  You et al. (2020) considers additive sparse (e.g., salt-and-pepper noise) noise only and proposes modeling the clean image and noise explicitly in the objective:

$$\min_{\boldsymbol{\theta},\boldsymbol{g},\boldsymbol{h}} \|\boldsymbol{y} - G_{\boldsymbol{\theta}}(\boldsymbol{z}) - (\boldsymbol{g}\circ\boldsymbol{g} - \boldsymbol{h}\circ\boldsymbol{h})\|_F^2, \tag{42}$$

where the overparametrized term $\boldsymbol{g}\circ\boldsymbol{g} - \boldsymbol{h}\circ\boldsymbol{h}$ ($\circ$ denotes the Hadamard product) is meant to capture the sparse noise, where a similar idea has proved effective for sparse recovery in Vaskevicius et al. (2019). Different properly-tuned learning rates for the clean image and sparse noise terms are necessary for success. The downside includes the prolonged running time as it pushes the peak reconstruction to the very last iteration, and the difficulty to extend the idea to other types of noise.

### A.7  ADDITIONAL EXPERIMENTAL DETAILS & RESULTS

#### A.7.1  EXTERNAL CODES

- DIP: `https://github.com/DmitryUlyanov/deep-image-prior`
- DD: `https://github.com/reinhardh/supplement_deep_decoder`
- DIP-TV: `https://github.com/sedaboni/ADMM-DIPTV`
- GP-DIP: `https://people.cs.umass.edu/~zezhoucheng/gp-dip/`
- DF-STE: `https://github.com/gistvision/dip-denosing`
- SV-ES: `https://github.com/sun-umn/Self-Validation`
- DOP: `https://github.com/ChongYou/robust-image-recovery`
- SB: `https://github.com/shizenglin/Measure-and-Control-Spectral-Bias`
- CBSD68: `https://github.com/clausmichele/CBSD68-dataset`

#### A.7.2  EXPERIMENT SETTINGS

Our default setup for all experiments is as follows. Our DIP model is the original one from Ulyanov et al. (2018); the optimizer is ADAM with a learning rate 0.01. For all other models, we use their default architectures, optimizers, and hyperparameters. For ES-WMV, the default window size $W = 100$, and patience number $P = 1000$. We use both PSNR and SSIM to access the reconstruction quality, and we report PSNR and SSIM gaps (the difference between our detected and peak numbers) as an indicator of our detection performance. **For most experiments, we repeat the experiments** 3 **times to report the mean and standard deviation**; when not, we explain why.

**Noise generation**  Following the noise generation rules of Hendrycks & Dietterich (2019)[1], we simulate four kinds of noise and three intensity levels for each noise type. The detailed information is as follows.

- **Gaussian noise:** 0 mean additive Gaussian noise with variance 0.12, 0.18, and 0.26 for low, medium, and high noise levels, respectively;

---

[1]`https://github.com/hendrycks/robustness`

- **Impulse noise:** also known as salt-and-pepper noise, replacing each pixel with probability $p \in [0, 1]$ into white or black pixel with half chance each. Low, medium, and high noise levels correspond to $p = 0.3, 0.5, 0.7$, respectively;

- **Speckle noise:** for each pixel $x \in [0, 1]$, the noisy pixel is $x(1 + \varepsilon)$, where $\varepsilon$ is 0-mean Gaussian with a variance level 0.20, 0.35, 0.45 for low, medium, and high noise levels, respectively;

- **Shot noise:** also known as Poisson noise. For each pixel, $x \in [0, 1]$, the noisy pixel is Poisson distributed with rate $\lambda x$, where $\lambda$ is $25, 12, 5$ for low, medium, and high noise levels, respectively.

### A.7.3  DENOISING EXAMPLES

On image denoising with different types and levels of noise, our ES method can help DIP to detect near-peak ES points, as shown in Fig. 11. We also explore the possibility of using the loss for ES here, but we fail to find correlations between the trend of the loss and that of the PSNR curve.

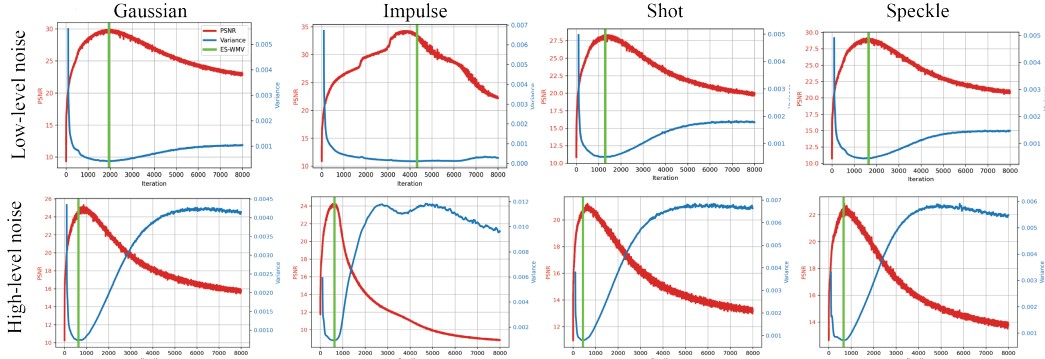

Figure 11: Our ES-WMV method on DIP for denoising "F16" with different noise types and levels (top: low-level noise; bottom: high-level noise). Red curves are PSNR curves, and blue curves are VAR curves. The green bars indicate the detected ES point.

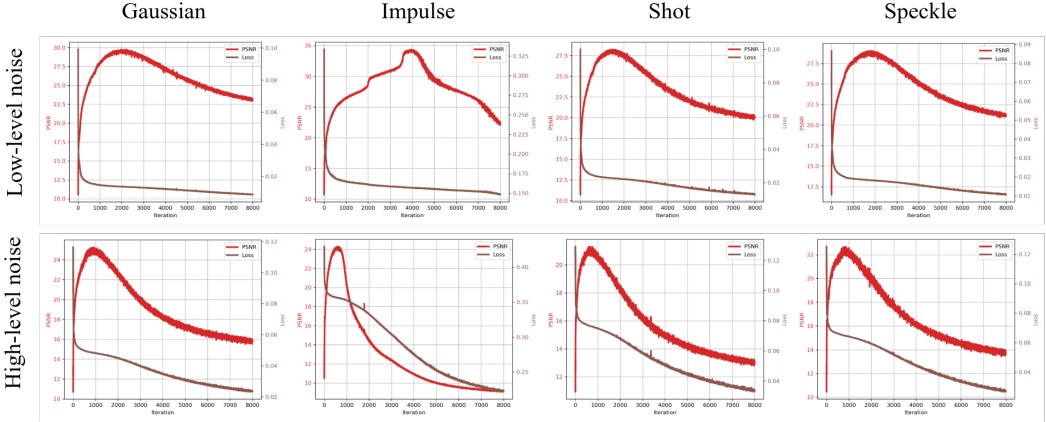

Figure 12: Our ES-WMV method on DIP for denoising "F16" with different noise types and levels (top: low-level noise; bottom: high-level noise). Red curves are PSNR curves, and brown curves are loss curves.

### A.7.4  COMPARISON WITH BASELINE METHODS

To further compare with baseline methods, we report the PSNR gaps of high-level noise cases and SSIM gaps of low- and high-level noise cases in Fig. 15, Fig. 16 and Fig. 17, respectively, which show a similar trend to the results of PSNR gaps. The detection gaps of our method are very marginal ($< 0.02$) for most noise types and levels (except for Baboon and Kodak1 for certain noise types/levels), while the baseline methods can well exceed 0.1 for most cases. In addition, we provide

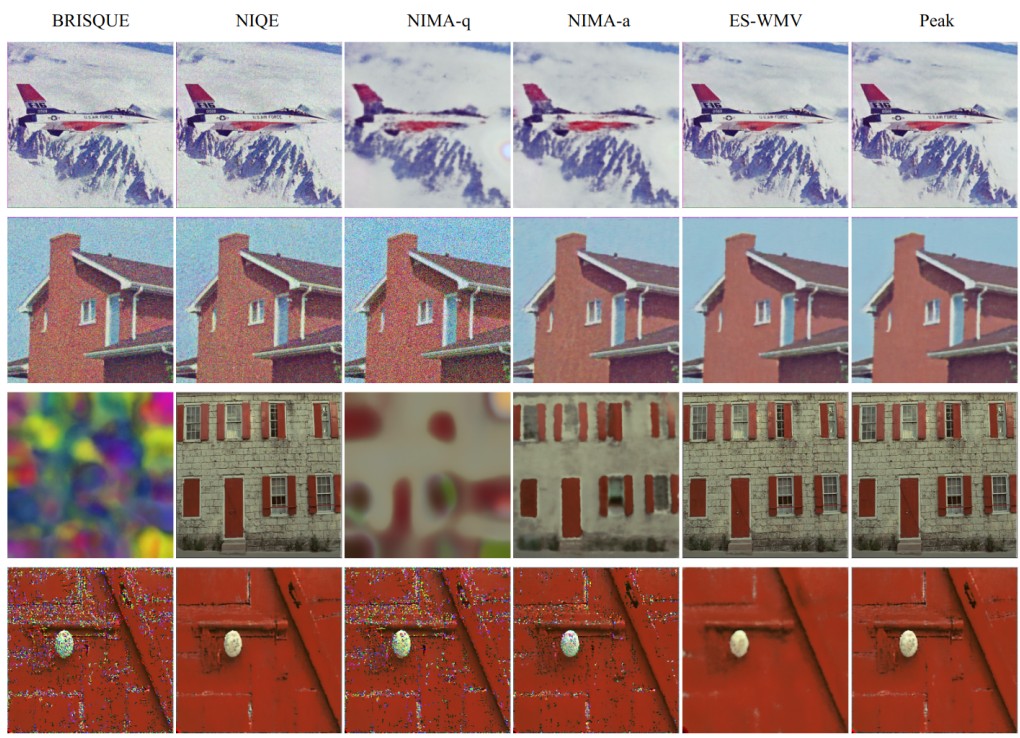

Figure 13: Visual comparisons of NR-IQMs and ES-WMV. From top to bottom: Gaussian noise (low), Gaussian noise (high), impulse noise (low), impulse noise (high).

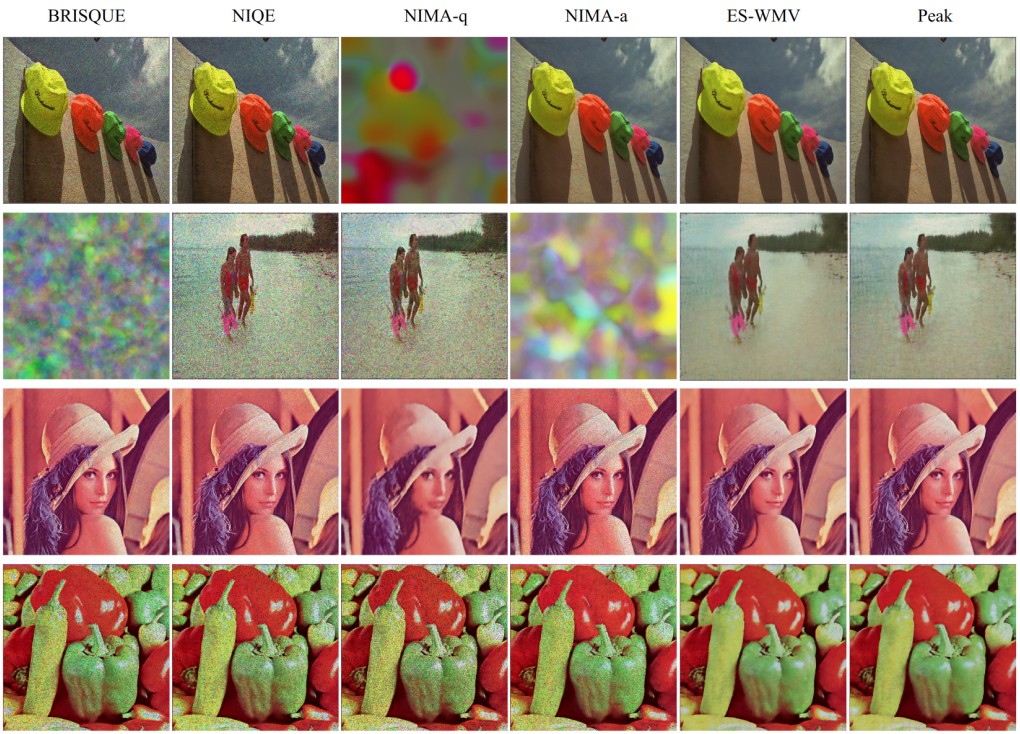

Figure 14: Visual comparisons of NR-IQMs and ES-WMV. From top to bottom: shot noise (low), shot noise (high), speckle noise (low), speckle noise (high).

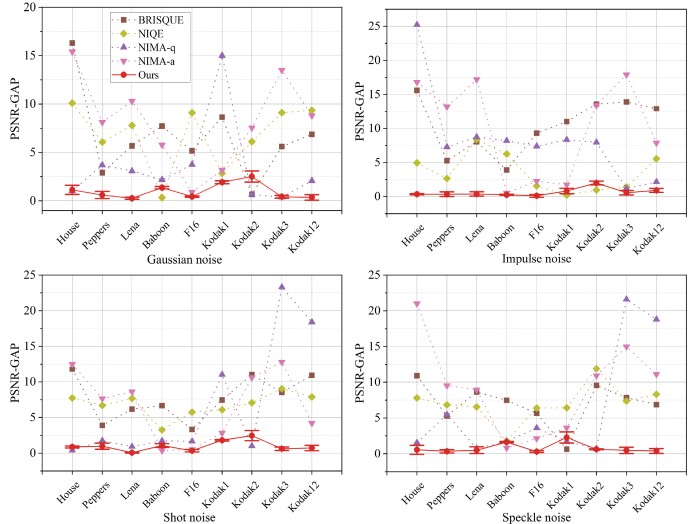

Figure 15: **High-level noise** detection performance in terms of PSNR gaps. For NIMA, we report both technical quality assessment (NIMA-q) and aesthetic assessment (NIMA-a). Smaller PSNR gaps are better.

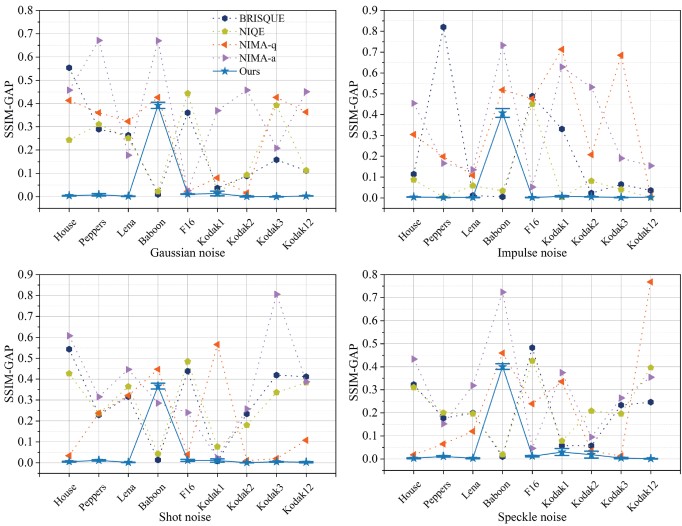

Figure 16: **Low-level noise** detection performance in terms of SSIM gaps. For NIMA, we report both technical quality assessment (NIMA-q) and aesthetic assessment (NIMA-a). Smaller SSIM gaps are better.

some visual detection results in Figs. 13 and 14. Our ES-WMV significantly outperforms than the four baseline methods visually.

### A.7.5 COMPARISON WITH COMPETING METHODS

Comparison between ES-WMV with DF-STE for Gaussian and shot noise on the 9-image dataset in terms of SSIM is reported in Fig. 18. Furthermore, we also test our ES-WMV and DF-STE on CBSD68 in Tab. 5. Our ES-WMV wins in high-level noise cases, but lags behind DF-STE in the low-level cases. The gaps between our ES-WMV and DF-STE for all noise levels mostly come from the peak-performance between the original DIP and DF-STE—modifications in DF-STE have affected the peak performance, positively for low-level cases and negatively for high-level cases, not

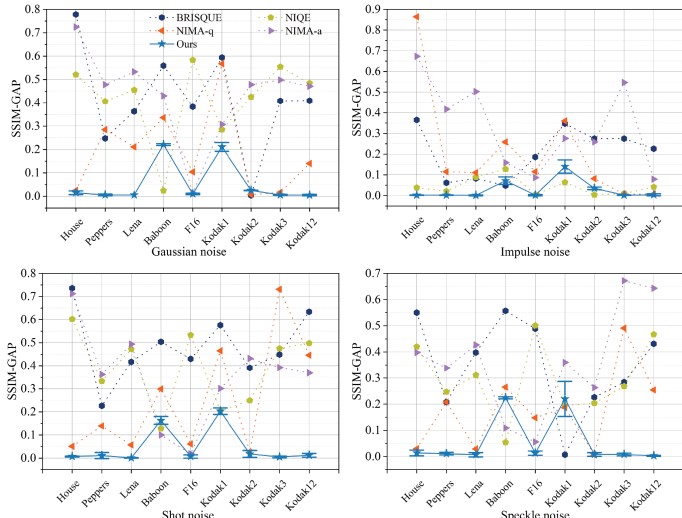

Figure 17: **High-level noise** detection performance in terms of SSIM gaps. For NIMA, we report both technical quality assessment (NIMA-q) and aesthetic assessment (NIMA-a). Smaller SSIM gaps are better.

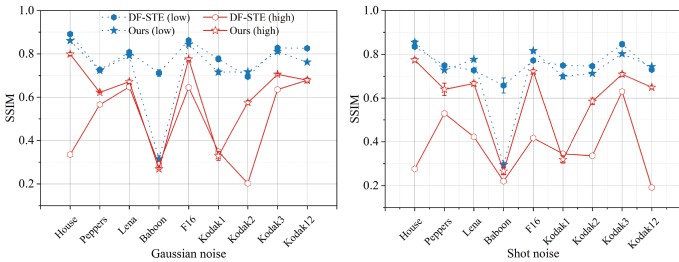

Figure 18: Comparison of DF-STE and ES-WMV for Gaussian and shot noise in terms of SSIM.

much from our ES method as evident from the uniformly small detection gaps reported in Tab. 5. Moreover, DF-STE can only handle Gaussian and Poisson noise for denoising, and the exact noise level is a required hyperparameter for their method to work.

Then we compare our ES-WMV and SV-ES in Fig. 19. The DIP results with ES-WMV vs. DOP on impulse noise are shown in Tab. 6. For SB, part of the qualitative detection results on the 9 images[2] is reported in Fig. 20.

Table 5: Comparison between ES-WMV and DF-STE for image denoising on the CBSD68 dataset with varying noise level $\sigma$: mean and (std). PSNR gaps below 1.0 are colored as red.

|  | $\sigma = 15$ | $\sigma = 25$ | $\sigma = 50$ |
|---|---|---|---|
| ES-WMV | 28.7(3.2) | 27.4(2.6) | 24.2(2.3) |
| DIP (Peak) | 29.7(3.0) | 28.0(2.4) | 24.9(2.3) |
| PSNR Gap | 1.0(0.7) | 0.7(0.5) | 0.7(0.5) |
| DF-STE | 31.4(1.8) | 28.4(2.2) | 21.1(2.5) |

### A.7.6 ES-WMV AS A HELPER

Performance of ES-WMV on DD, GP-DIP, DIP-TV, and SIREN for Gaussian denoising in terms of SSIM gaps (see Fig. 21).

---

[2]http://www.cs.tut.fi/~foi/GCF-BM3D/index.html#ref_results

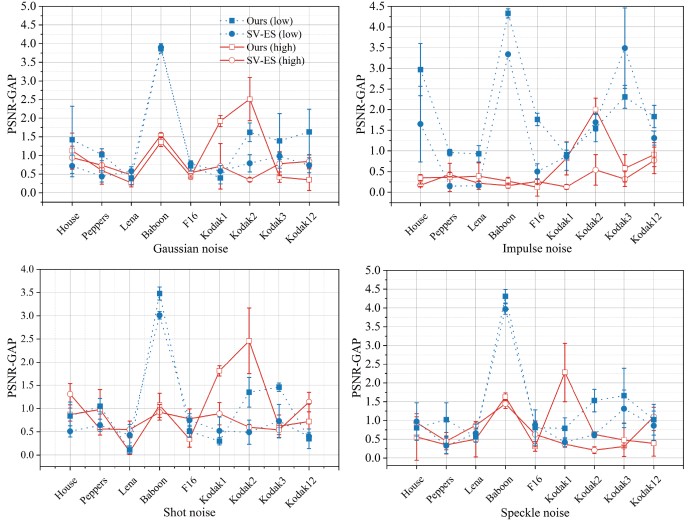

Figure 19: **Low- and high-level noise** detection performance of SV-ES and ours in terms of PSNR gaps.

Table 6: DIP with ES-WMV vs. DOP on impulse noise: mean and (std).

|  | Low Level | | High Level | |
|---|---|---|---|---|
|  | PSNR | SSIM | PSNR | SSIM |
| DIP-ES | 31.64 (5.69) | 0.85 (0.18) | 24.74 (3.23) | 0.67 (0.19) |
| DOP | 32.12 (4.52) | 0.92 (0.07) | 27.34 (3.78) | 0.86 (0.10) |

### A.7.7 PERFORMANCE ON REAL-WORLD DENOISING

Table 7: DIP with ES-WMV on real image denoising on the PolyU Dataset: mean and (std). (**D**: Detected)

|  | PSNR(**D**) | PSNR Gap | SSIM(**D**) | SSIM Gap |
|---|---|---|---|---|
| DIP (MSE) | 36.83 (3.07) | 1.26 (1.22) | 0.98 (0.02) | 0.01 (0.01) |
| DIP ($\ell_1$) | 36.20 (2.81) | 1.64 (1.58) | 0.97 (0.02) | 0.01 (0.01) |
| DIP (Huber) | 36.76 (2.96) | 1.28 (1.09) | 0.98 (0.02) | 0.01 (0.01) |

As stated from the beginning, ES-WMV is designed with real-world IPs, targeting unknown noise types and levels. Given the encouraging performance above, we test it on a common real-world denoising dataset—PolyU Dataset Xu et al. (2018), which contains 100 cropped regions of $512 \times 512$ from 40 scenes. The results are reported in Tab. 7. We do not repeat the experiments here; the means and standard deviations are obtained over the 100 images of the PolyU dataset. On average, our detection gaps are $\leq 1.64$ in PSNR and $\leq 0.01$ in SSIM for this dataset across various losses. The absolute PNSR and SSIM detected are surprisingly high.

### A.7.8 RESULTS FOR MRI RECONSTRUCTION

The detection performance of ES-WMV for MRI reconstruction is shown in Fig. 23 in terms of SSIM.

### A.7.9 IMAGE INPAINTING

In this task, a clean image $\boldsymbol{x}_0 \in [0, 1]^{H \times W}$ is contaminated by additive Gaussian noise $\varepsilon$, and then only partially observed to yield the observation $\boldsymbol{y} = (\boldsymbol{x}_0 + \boldsymbol{\varepsilon}) \odot \boldsymbol{m}$, where $\boldsymbol{m} \in \{0, 1\}^{H \times W}$ is a binary mask and $\odot$ denotes the Hadamard product. Given $\boldsymbol{y}$ and $\boldsymbol{m}$, the goal is to reconstruct $\boldsymbol{x}_0$. We consider the formulation reparametrized by DIP, where $G_{\boldsymbol{\theta}}$ is a trainable DNN parametrized by

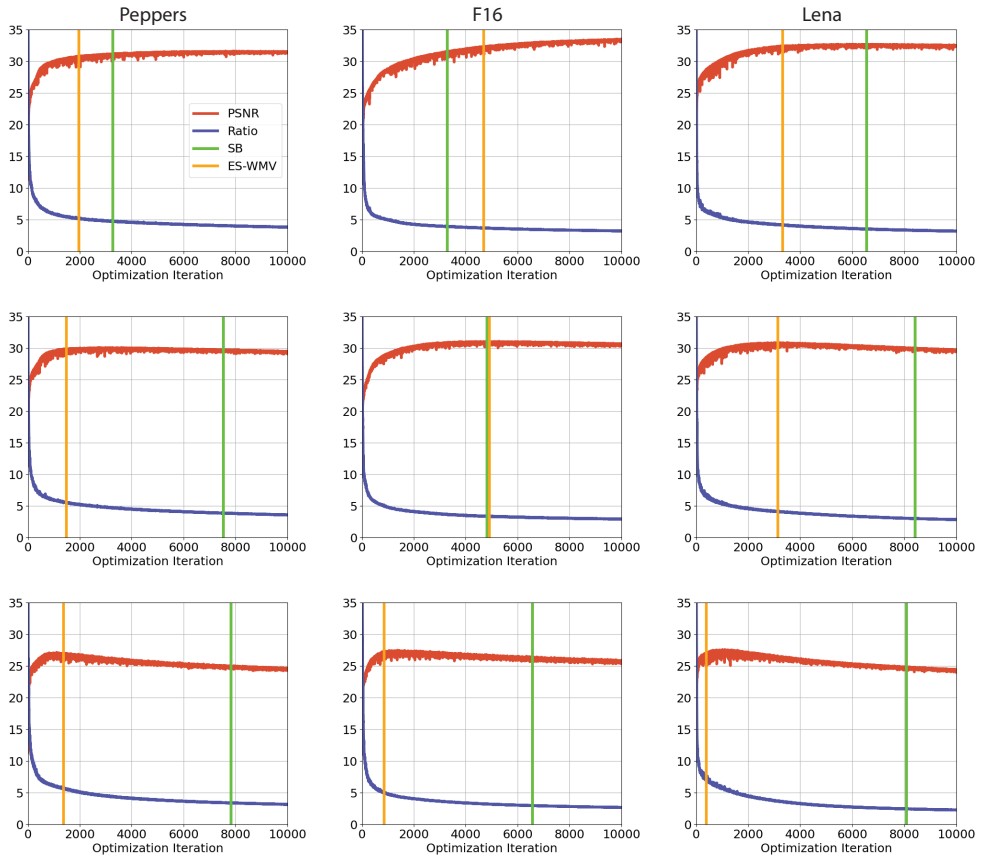

Figure 20: Comparison between ES-WMV and SB for image denoising (top: $\sigma = 15$; middle: $\sigma = 25$; bottom: $\sigma = 50$). The red and blue curves are the PNSR and the ratio metric curves. The orange and green bars indicate the ES points detected by our ES-WMV and SB, respectively.

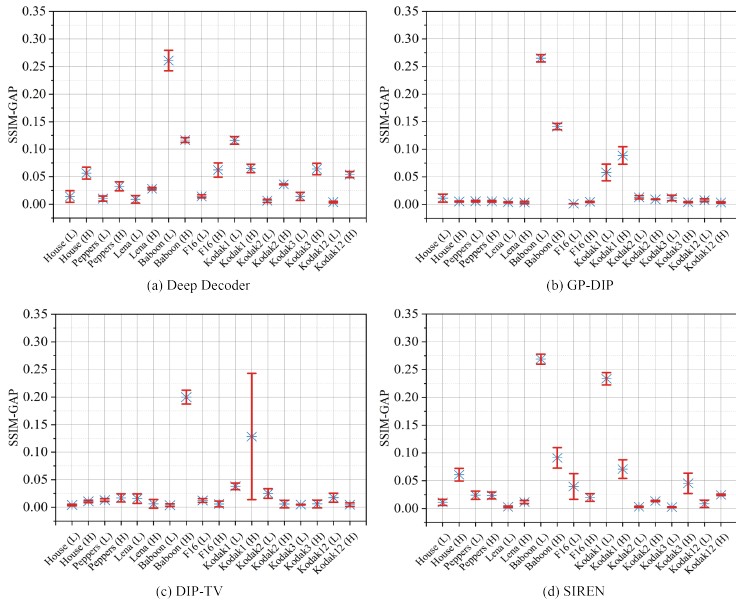

Figure 21: Performance of ES-WMV on DD, GP-DIP, DIP-TV, and SIREN for Gaussian denoising in terms of SSIM gaps. L: low noise level; H: high noise level

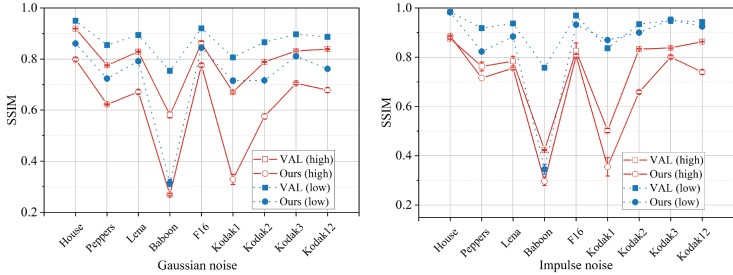

Figure 22: Comparison of VAL and ES-WMV for Gaussian and impulse noise in terms of SSIM.

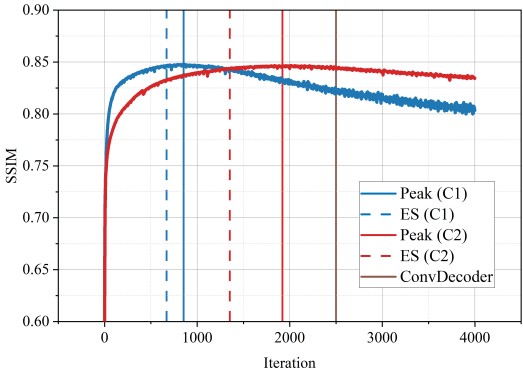

Figure 23: Detection on MRI reconstruction

$\boldsymbol{\theta}$ and $\boldsymbol{z}$ is a frozen random seed:

$$\ell(\boldsymbol{\theta}) = \|(G_{\boldsymbol{\theta}}(\boldsymbol{z}) - \boldsymbol{y}) \odot \boldsymbol{m}\|_F^2. \tag{43}$$

The mask $\boldsymbol{m}$ is generated according to an iid Bernoulli model with a rate of $50\%$, i.e., half of pixels not observed in expectation. The **noise $\varepsilon$ is set to the medium level**, i.e., additive Gaussian with $0$ mean and $0.18$ variance. We test our ES-WMV for DIP on the inpainting dataset used in the original DIP paper Ulyanov et al. (2018). The PSNR gaps are $\leq 1.00$ and the SSIM gaps are $\leq 0.05$ for most cases (see Tab. 8). We also visualize two examples in Fig. 24.

Table 8: Detection performance of DIP with ES-WMV for image inpainting: mean and (std). PSNR gaps below 1.00 are colored as red; SSIM gaps below 0.05 are colored as blue. (**D**: Detected)

|  | PSNR(**D**) | PSNR Gap | SSIM(**D**) | SSIM Gap |
|---|---|---|---|---|
| Barbara | 21.59 (0.03) | 0.20 (0.03) | 0.67 (0.00) | 0.00 (0.00) |
| Boat | 21.91 (0.10) | 1.16 (0.18) | 0.68 (0.00) | 0.03 (0.01) |
| House | 27.95 (0.33) | 0.48 (0.10) | 0.89 (0.01) | 0.01 (0.00) |
| Lena | 24.71 (0.30) | 0.37 (0.18) | 0.80 (0.00) | 0.01 (0.00) |
| Peppers | 25.86 (0.22) | 0.23 (0.05) | 0.84 (0.01) | 0.02 (0.00) |
| C.man | 25.26 (0.09) | 0.23 (0.14) | 0.82 (0.00) | 0.01 (0.00) |
| Couple | 21.40 (0.44) | 1.21 (0.53) | 0.63 (0.01) | 0.04 (0.02) |
| Finger | 20.87 (0.04) | 0.24 (0.17) | 0.77 (0.00) | 0.01 (0.01) |
| Hill | 23.54 (0.08) | 0.25 (0.11) | 0.70 (0.00) | 0.00 (0.00) |
| Man | 22.92 (0.25) | 0.46 (0.11) | 0.70 (0.01) | 0.01 (0.00) |
| Montage | 26.16 (0.33) | 0.38 (0.26) | 0.86 (0.01) | 0.03 (0.01) |

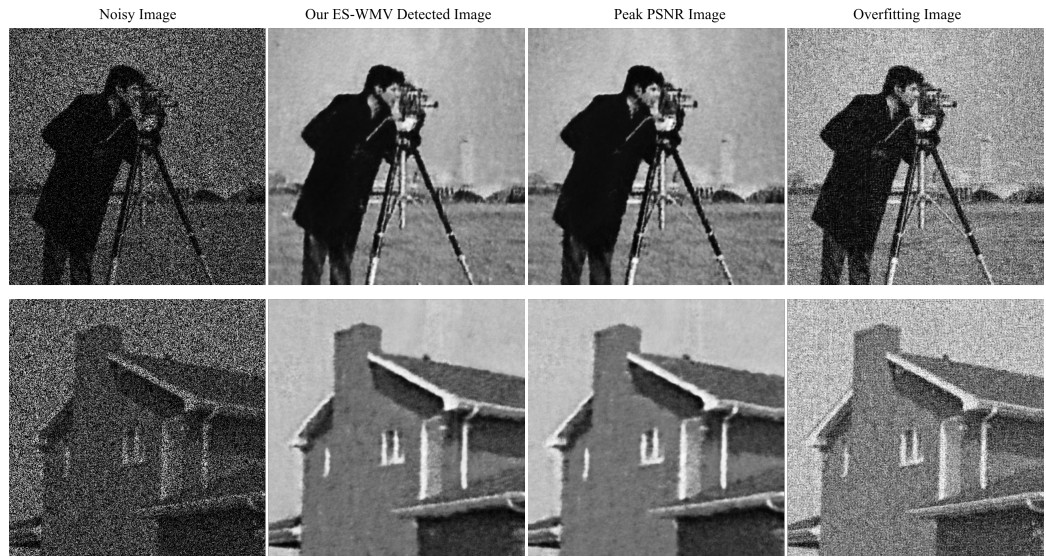

Figure 24: Visual detection performance of ES-WMV on image inpainting.

### A.7.10    IMAGE SUPER-RESOLUTION

In this task, a degraded observation $y$ is obtained as the downsampled version of a noisy image: i.e., $y = \mathcal{D}_t(x_0 + \varepsilon)$, where $\mathcal{D}_t(\cdot) : [0, 1]^{3 \times tH \times tW} \to [0, 1]^{3 \times H \times W}$ is a *downsampling operator* that resizes an image by the factor $t$. Then given $y$ and $t$, the goal is to reconstruct $x_0$. We consider the formulation reparametrized by DIP, where $G_\theta$ is a trainable DNN parametrized by $\theta$ and $z$ is a frozen random seed:

$$\ell(\boldsymbol{\theta}) = \|\mathcal{D}_t(G_{\boldsymbol{\theta}}(\boldsymbol{z})) - \boldsymbol{y}\|_F^2. \tag{44}$$

The **noise $\varepsilon$ is again set to the medium level**, i.e., additive Gaussian with $0$ mean and $0.18$ variance. We test our ES-WMV for DIP on the super-resolution dataset used in the original DIP paper Ulyanov et al. (2018). The PSNR gaps are $\leq 1.00$ and the SSIM gaps are $\leq 0.05$ for most cases (see Tab. 9). Our ES-WMV is again able to detect near-peak performance for most images.

Table 9: Detection performance of DIP with ES-WMV for $4\times$ **image super-resolution**: mean and (std). PSNR gaps below $1.00$ are colored as red; SSIM gaps below $0.05$ are colored as blue. (**D**: Detected)

|  | PSNR(**D**) | PSNR Gap | SSIM(**D**) | SSIM Gap |
|---|---|---|---|---|
| Baboon | 17.82 (0.02) | 0.10 (0.04) | 0.38 (0.00) | 0.01 (0.01) |
| Barbara | 19.93 (0.05) | 0.04 (0.01) | 0.59 (0.01) | 0.01 (0.00) |
| Bridge | 18.04 (0.04) | 0.33 (0.09) | 0.43 (0.00) | 0.00 (0.00) |
| Coastguard | 20.76 (0.05) | 0.17 (0.13) | 0.53 (0.01) | 0.02 (0.01) |
| Comic | 16.70 (0.07) | 0.06 (0.06) | 0.45 (0.01) | 0.00 (0.00) |
| Face | 21.67 (0.12) | 0.63 (0.12) | 0.56 (0.01) | 0.06 (0.01) |
| Flowers | 18.96 (0.08) | 0.12 (0.03) | 0.56 (0.01) | 0.02 (0.00) |
| Foreman | 20.62 (0.04) | 0.35 (0.07) | 0.69 (0.00) | 0.06 (0.00) |
| Lena | 22.40 (0.07) | 0.30 (0.08) | 0.70 (0.00) | 0.04 (0.00) |
| Man | 19.94 (0.07) | 0.22 (0.05) | 0.52 (0.00) | 0.02 (0.01) |
| Monarch | 19.68 (0.90) | 1.40 (0.90) | 0.72 (0.00) | 0.03 (0.00) |
| Pepper | 21.20 (0.14) | 0.14 (0.04) | 0.67 (0.01) | 0.04 (0.01) |
| Ppt3 | 17.55 (0.10) | 0.19 (0.10) | 0.71 (0.01) | 0.01 (0.00) |
| Zebra | 19.09 (0.08) | 0.10 (0.05) | 0.56 (0.01) | 0.01 (0.01) |

### A.7.11 ES-WMV vs. ES-EMV

We now consider our memory-efficient version (ES-EMV) as described in Algorithm 2, and compare it with ES-WMV, as shown in Fig. 25. Besides the memory benefit, ES-EMV runs around 100 times faster than ES-WMV, as reported in Tab. 2 and does seem to provide a consistent improvement on the detected PSNRs for image denoising tasks on NTIRE 2020 Real Image Denoising Challenge (Abdelhamed et al., 2020), PolyU dataset Xu et al. (2018) and the classic 9-image dataset (Dabov et al., 2008) (see Tabs. 10 and 11 and Fig. 25), due to the strong smoothing effect (we set $\alpha = 0.1$). In this paper, we prefer to stay simple and leave systematic evaluations of these variants for more tasks as future work.

Table 10: Detection performance comparison between DIP with ES-WMV and DIP with ES-EMV for real image denoising on $1024$ images from the RGB track of NTIRE 2020 Real Image Denoising Challenge (Abdelhamed et al., 2020): mean and (std). Higher PSNR and SSIM are in red. (**D**: Detected)

|  | PSNR(**D**)-WMV | PSNR(**D**)-EMV | SSIM(**D**)-WMV | SSIM(**D**)-EMV |
|---|---|---|---|---|
| DIP (MSE) | 34.04 (3.68) | 34.96 (3.80) | 0.92 (0.07) | 0.93 (0.07) |
| DIP ($\ell_1$) | 33.92 (4.34) | 34.83 (4.35) | 0.93 (0.05) | 0.94 (0.05) |
| DIP (Huber) | 33.72 (3.86) | 34.72 (4.04) | 0.92 (0.06) | 0.93 (0.06) |

Table 11: Detection performance comparison between DIP with ES-WMV and DIP with ES-EMV for real image denoising on the PolyU dataset Xu et al. (2018): mean and (std). Higher PSNR and SSIM are in red. (**D**: Detected)

|  | PSNR(**D**)-WMV | PSNR(**D**)-EMV | SSIM(**D**)-WMV | SSIM(**D**)-EMV |
|---|---|---|---|---|
| DIP (MSE) | 36.83 (3.07) | 37.32 (3.82) | 0.98 (0.02) | 0.98 (0.03) |
| DIP ($\ell_1$) | 36.20 (2.81) | 36.43 (3.22) | 0.97 (0.02) | 0.97 (0.02) |
| DIP (Huber) | 36.76 (2.96) | 37.21 (3.19) | 0.98 (0.02) | 0.98 (0.02) |

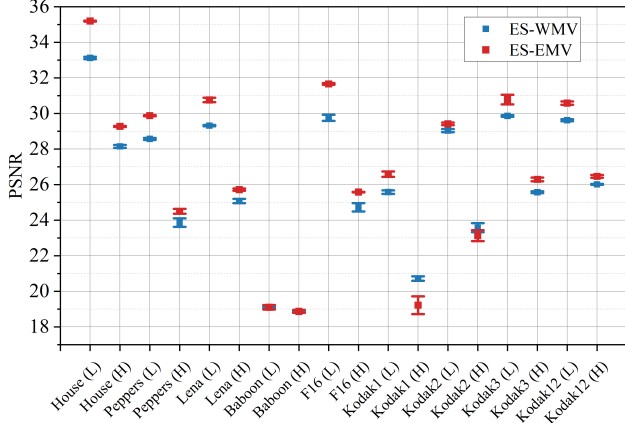

Figure 25: Detected PSNR comparison between DIP with ES-WMV and DIP with ES-EMV on the classic 9-image dataset (Dabov et al., 2008).

### A.7.12 BLIND IMAGE DEBLURRING (BID)

In this section, we systematically test our ES-WMV and VAL on the entire standard Levin dataset for both low-level and high-level cases. We set the maximum number of iterations as $10,000$ to ensure that we perform sufficient optimization. The detected images of our ES-WMV are substantially better than those of VAL, as shown in Tab. 12.

Table 12: BID detection comparison between ES-WMV and VAL on the Levin dataset for both low-level and high-level noise: mean and (std).Higher PSNR is in red and higher SSIM is in blue. (**D**: Detected)

| | Low Level | | High Level | |
|---|---|---|---|---|
| | PSNR(**D**) | SSIM(**D**) | PSNR(**D**) | SSIM(**D**) |
| WMV | 28.54(0.61) | 0.83(0.04) | 26.41(0.67) | 0.76(0.04) |
| VAL | 18.87(1.44) | 0.50(0.09) | 16.69(1.39) | 0.44(0.10) |

### A.7.13 ABLATION STUDY

We vary the window size $W$ (default 100) and patience number $P$ (default: 1000) across a range and check how the detection gap changes for Gaussian denoising with medium-level noise on the classic 9-image dataset (see:Fig. 26).

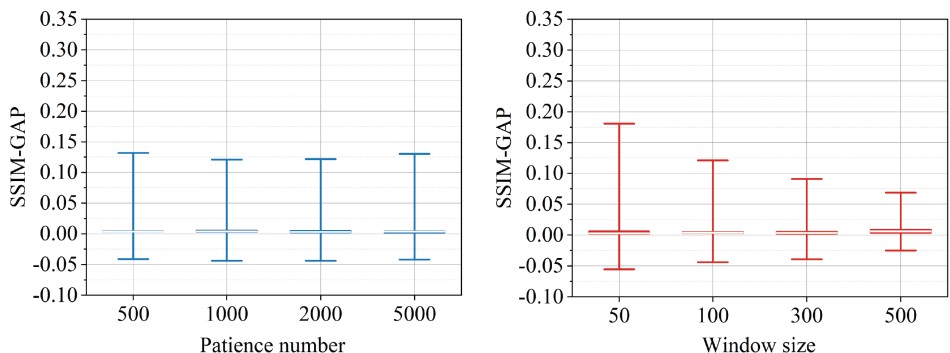

Figure 26: Effect of patience number and window size on detection in terms of SSIM gaps

### A.8 ANALYSIS OF FAILURE CASES IN FIG. 8

We note that there are some occasional failures cases when applying our ES on some DIP variants in Fig. 8. In this section, we provide VAR curves of these cases. For the failure of GP-DIP on the "House (L)" image in Fig. 8, GP-DIP has a weird multi-valley, gradual descending pattern in the VAR curve, corresponding to a multi-peak, gradual ascending pattern in the PSNR curve. The first major valley in the VAR curve is roughly aligned with the first major peak, not the final best peak, in the PSNR curve. So although our valley-detection method successfully detects the first major valley, the PSNR gap is relatively large. Overall, although our ES method works well with GP-DIP for most of the test cases, we would not recommend GP-DIP for practical use. The concern is the speed: as a method trying to mitigate the overfitting, the best reconstruction of GP-DIP tends to be around the very last iterates. The failure on the "Lena(L)" image is due to a similar multivalley pattern in the VAR curve.

For both cases, we observe that using smaller learning rates for GP-DIP and DD helps to smooth out their curves and mitigate the multi-valley phenomenon which likely will lead to much smaller detection gaps. We hesitate to refine in this direction, as our focus of this paper is on the ES method itself.

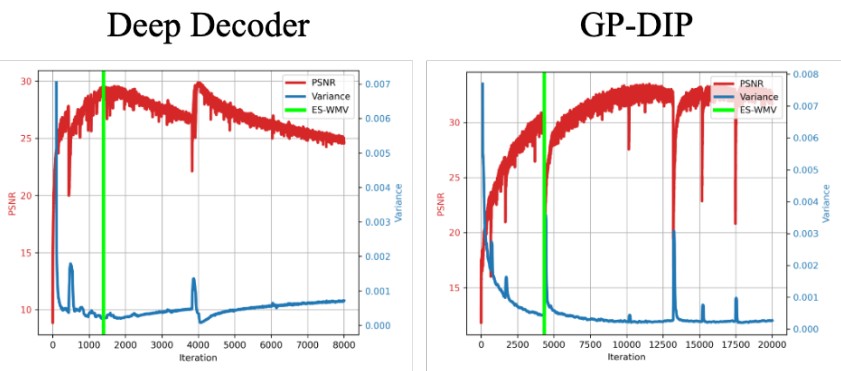

Figure 27: VAR curves of failure cases. Left: DD for "Lena(L)"; Right: GP-DIP for "House(L)" in Fig. 8.

