# OpenReview forum: "Early Stopping for Deep Image Prior"
_ICLR.cc/2023/Conference — Submitted to ICLR 2023_

### Official Review · Reviewer_kTJ2 · 2022-10-24

**Confidence:** 4
**Correctness:** 3
**Technical Novelty And Significance:** 2
**Empirical Novelty And Significance:** 3
**Recommendation:** 6

**Clarity, Quality, Novelty And Reproducibility:**

Clarity: the paper is easy to follow and clear in most parts.
Quality: I have some concerns with respect to the empirical and theoretical support of the proposed method (see Weaknesses above).
Novelty: to the best of my knowledge the proposed early stopping scheme is novel for Deep Image Prior.
Reproducibility: code is provided to reproduce the experimental results.

Minor suggestions:

- Typo in 'Prior work addressing the overfitting': 'noisy type and level'

- Typo in 'Intuition for our method': 'more details...'

- Possible typo in Theorem 2.1: what does index m denote in C_{m, \eta, \sigma_i}? It should probably be W.

- In Algorithm 2 in the appendix VAR is not defined.

**Strength And Weaknesses:**

Strengths:

-  The paper is very well-written and organized. It is easy to follow and the structure is logical.

- The problem the paper is investigating is a very relevant one and therefore this paper is well-motivated.

- The proposed technique can be combined with any DIP-based method, is simple to implement and the overhead in terms of computation is minimal.  Therefore, it has potential in practice.

Weaknesses:

- I am not completely satisfied with the justification of tracking running variance to detect peak performance. The theoretical justification doesn't seem to be too convincing, as the provided bounds are very loose, and having a U-shaped upper bound does not necessarily mean that the running variance will also be U-shaped. Even though the experiments show that the metric works in some cases, it is not clear to me why variance should be minimized close to the solution and increase during overfitting.

- Based on the experimental results, the proposed technique is somewhat inaccurate in low-level noise scenarios (Figure 3 top row) and often surpassed by other methods (Table 3, Figure 6, Figure 7). A more in-depth study on the noise regime where the proposed method is reliable would be very useful.

- The proposed method has a significant memory overhead. Storing W=100 iterates in some applications may be prohibitive. The proposed EMV direction is interesting and can potentially address this issue, however currently there are not enough experiments on this variant to fully support its merit.

- Most experiments are on very small datasets (1-9 images). Evaluation on larger datasets would help filtering out statistical variations in the presented results.

**Summary Of The Paper:**

Authors propose a technique to detect peak performance without a reference for Deep Image Prior (DIP) based computational imaging algorithms.  The proposed strategy utilizes the running variance of past reconstruction iterates to determine when to terminate training. Authors provide theoretical results to justify the proposed early stopping criterion. Through numerical experiments, they demonstrate that the proposed method can detect peak performance reasonably well across a wide range of inverse problems and types of measurement noise.

**Summary Of The Review:**

Overall, in its current form I would recommend borderline rejecting the paper. The proposed method shows promise and works reasonably well across different types of additive noise. However, in my opinion the justification for the particular early stopping metric is somewhat lacking. The provided theory doesn't provide a satisfying answer either. Could the authors provide more explanation and further backing why running variance would show the U-shape phenomenon close to the ground truth signal? Furthermore,  the experimental results seem to show that the proposed technique is mostly reliable in high-noise scenario and a detailed investigation with respect to noise level limitations is missing. Could the authors provide more discussion on the aforementioned limitations?

---

> ### Author Response · Authors · 2022-11-19
> **Response to Reviewer kTJ2 (1/2)**
>
> We thank reviewer kTJ2 for the detailed review and insightful comments.
> ## Q1. I am not completely satisfied with the justification of tracking running variance to detect peak performance. The theoretical justification doesn't seem to be too convincing, as the provided bounds are very loose, and having a U-shaped upper bound does not necessarily mean that the running variance will also be U-shaped. Even though the experiments show that the metric works in some cases, it is not clear to me why variance should be minimized close to the solution and increase during overfitting.
>
> Thank you for raising this concern! We position this paper mostly as an empirical paper, and our ES method has been motivated by empirical observations on the variance curve. We agree that our theoretical development, as a first attempt, is incomplete—we also do not claim the theoretical development as among our main contributions.
>
> The U-shape upper bound in Theorem 2.2 does leave considerable gaps there. But, we would also like to point out that many theoretical results especially for deep neural networks including optimization, generalization, and many other aspects are using mathematical upper bounds for verifying the behaviors in practice. In particular, two previous theoretical papers cited below (also cited in our paper [1,2]) that try to prove the bell-shaped learning curve of DIP also only provide upper bounds of similar nature.
>
> >[1] Heckel, R. and Soltanolkotabi, M., 2020, November. Compressive sensing with un-trained neural networks: Gradient descent finds a smooth approximation. In International Conference on Machine Learning (pp. 4149-4158). PMLR.
>
> >[2] Heckel, R. and Soltanolkotabi, M., 2019. Denoising and regularization via exploiting the structural bias of convolutional generators. arXiv preprint arXiv:1910.14634.
>
> ## Q2. Based on the experimental results, the proposed technique is somewhat inaccurate in low-level noise scenarios (Figure 3 top row) and often surpassed by other methods (Table 3, Figure 6, Figure 7). A more in-depth study on the noise regime where the proposed method is reliable would be very useful.
>
> Thank you for raising this point! We agree that our ES method is not always the best one, although in the majority of cases it is. But our method wins in generality and overall strong results:
>
> * DF-STE only handles Gaussian and Poisson noise for denoising, and noise level needs to be a hyperparameter for their method to work;
> * SB paper consists of a combination of a DIP variant and an ES detection method. Their ES detection method only works for their DIP variant, as noted in the original SB paper ``Note that the stopping criterion fails for the original deep image prior".
> * VAL works strong for denoising, but they produce unacceptable results on blind deblurring.
>
> So it is a bit unfair to access our method only from noise levels; we argue that one needs to take into account all these factors: noise type and level, task, and DIP variant when accessing all the methods under comparison here.
>
> Please also refer to our General R2 (in general response) regarding the concern for our ES when facing low-level noise.
>
> ## Q3. The proposed method has a significant memory overhead. Storing W=100 iterates in some applications may be prohibitive. The proposed EMV direction is interesting and can potentially address this issue, however, currently there are not enough experiments on this variant to fully support its merit.
>
> Thanks for raising this insightful point!  We have conducted more experiments to compare the two variants of our ES method, i.e., ES-WMV and ES-EMV, in Appendix A.7.11. ES-EMV seems to provide a consistent improvement on the detected PSNRs for image denoising tasks on several datasets due to the strong smoothing effect (we set α = 0.1). In this paper, we prefer to stay simple and leave systematic evaluations of these variants as future work.
>
> ## Q4. Most experiments are on very small datasets (1-9 images). Evaluation on larger datasets would help filtering out statistical variations in the presented results.
>
> Thanks for raising the point! We totally agree with the thought; we have two large-scale datasets for denoising, i.e.,  NTIRE 2020 Real Image Denoising Challenge in Table 1 and the PolyU Dataset in Table 6, and also in this revision, we have expanded our evaluation of blind deblurring to the whole Levin dataset, a gold-standard to blind deblurring evaluation. We do hope to do more thorough experiments in the future if we specialize our methods to any particular applications, but currently, as a methodology paper, we have to strike a balance between the thoroughness of evaluation on each task and the range of tasks we cover.

---

> > ### Author Response · Authors · 2022-11-19
> > **Response to Reviewer kTJ2 (2/2)**
> >
> > ## Q5. Typo in 'Prior work addressing the overfitting': 'noisy type and level'
> >
> > Thanks for the suggestion. We have corrected it in the updated submission.
> >
> > ## Q6. Typo in 'Intuition for our method': 'more details...'
> >
> > Thanks for the suggestion. We have corrected it in the updated submission.
> >
> > ## Q7. Possible typo in Theorem 2.1: what does index m denote in C_{m, \eta, \sigma_i}? It should probably be W.
> >
> > Thanks for the suggestion. Yes, it should be W and we have corrected it in the updated submission.
> >
> > ## Q8. In Algorithm 2 in the appendix VAR is not defined.
> >
> > Thanks for the suggestion. VAR should be EMV and we have corrected it in the updated submission.

---

> ### Author Response · Authors · 2022-12-05
> **We are anticipating your post-rebuttal feedback!**
>
> Dear Reviewer kTJ2,
>
> We would like to sincerely thank you again for your time in reviewing our work!
>
> We understand you might be quite busy. However, as the discussion deadline is approaching, would you mind checking our response and confirming whether you have any further questions? Any further comments are discussions are welcomed!
>
> Best Regards,
>
> The authors of Paper2037

---

> ### Author Response · Authors · 2022-12-08
> **A kind reminder: Deadline of the final discussion stage (12/12) is fast approaching!**
>
> Dear reviewer kTJ2,
>
> We would like to sincerely thank you again for your time and efforts for your insightful reviews!
>
> We are so sorry for keeping reminding you. We are totally okay if your final decision is to keep the score. We are writing simply to remind you that we have made the clarification and experiments as you suggested. Do they make sense to you? And possibly, do they change your mind a little bit?
>
> We value the feedback much more than a sole score. So we would really appreciate it if you could give us any feedback (like, if you think our point is not convincing, what kinds of experiments you think are missing to support the claim?). Your opinions are rather important for us to improve the work! Thanks again!
>
> Best Regards,
>
> The authors of Paper2037

---

> > ### Comment · Reviewer_kTJ2 · 2022-12-09
> > **Response to authors**
> >
> > Thank you for addressing my concerns in detail and I apologize for the belated reply. I understand that the paper is positioned more as an empirical paper and therefore the purpose of the theory work is to provide some grounding and intuition. Overall, I still believe that more careful ablation studies on the limitations of the method would be necessary, as the focus is on practical deployment. However, I raise my score to 6 as most of my other concerns have been addressed and I think there is merit to this work.

---

> > > ### Author Response · Authors · 2022-12-10
> > > **Re: Response to authors**
> > >
> > > Dear Reviewer kTJ2,
> > >
> > > We sincerely thank you for your valuable response and increasing the score! Actually, our General R2 (in general response) further explores the limitation concern and possible refinement for our ES when facing low-level noise, but we also plan to conduct more careful ablation studies on the limitations of our ES as you suggested in the future modified version.
> > >
> > > Best Regards,
> > >
> > > The authors of Paper2037

---

### Official Review · Reviewer_sEH9 · 2022-10-25

**Confidence:** 4
**Correctness:** 3
**Technical Novelty And Significance:** 3
**Empirical Novelty And Significance:** 3
**Recommendation:** 6

**Clarity, Quality, Novelty And Reproducibility:**

- Concern regarding the significance of the results:

    1) One aspect of measuring the significance of a work is how it performs against the SOTA methods. However, this paper has only focused on the performance gain over the solutions for the family of DIP and ignored considering SOTA methods in general, either supervised or unsupervised.

- Clarification questions:

    2) For Table 2, it is not clear if the wall-clock time of the three ES methods should be added to DIP’s to achieve the total wall-clock time per iteration? Or that ES-WMV has some optimization resulting in lower wall-clock time per iteration when used with DIP, compared to only using DIP?

    - For the experiment on image deblurring:

        3) How are the overfit iteration numbers chosen?

        4) Why only a subset of images/kernels are chosen, and what was the criterion to choose them?
- Minor point:

    5) The references are not written in a consistent manner. For example, few times the full name of a conference is written and at other times its abbreviation such as NeurIPS. Also, sometimes the URL and/or DOI are provided and sometimes they are not.

**Strength And Weaknesses:**

- S1: The motivation and justification of using the proposed metric for ES is discussed in details.

- S2: Providing extensive experiments on denoising and showing robustness to different noise types and higher intensities of noise.

- S3: Outperforming non-reference based image quality metrics and other methods of ES for denoising specifically designed for DIP either in terms of average quality or wall-clock time per iteration

- S4: Proposing a lighter variant of the algorithm to save on computations per iteration

- W1: The method does not perform well when an image consists of too much high-ferequency components.

- W2: Based on Figures 3, 8a-b, and 17, it seems like in practice the method is prone to higher error or failure when the changes in the PSNR curve are relatively smooth or when the noise level is low.

- W3: The experiments are imbalanced and mainly focused on denoising. For example, for blind image deblurring a limited subset of the dataset and/or kernels are considered and the the results in case of the default noise levels used in the original Ren et al. (2020) solution are not provided.

**Summary Of The Paper:**

The paper proposes an Early Stopping (ES) method for Deep Image Prior (DIP) based on the running variance of the intermediate outputs. For the task of denoising, a broad set of experiments is provided to study it under different noise types and intensities and against other approaches for early stopping. Besides, some experiments showing how it can be utilized in blind image deblurring and MRI reconstruction are provided.

**Summary Of The Review:**

The empirical results seems to be well-aligned with the justification of the proposed method. The performance is generally improved on denoising compared to previous ES for DIP methods and the experiments on this task are extensive. There are some situations such as too much high-frequency components or too much smoothness in the optimization curve that could affect the result negatively. While the idea of tracking the running variance of the outputs seems to be simple, I have not seen it used for ES in DIP. The significance cannot be properly measured when only looking at family of DIP methods and not including other SOTA methods for denoising. The situation is worse on the other tasks as the comparisons are far more limited and does not even include latest methods based on DIP, for example for deblurring, not to mention the rest of SOTA methods.

---

> ### Author Response · Authors · 2022-11-19
> **Response to Reviewer sEH9**
>
> We thank reviewer sEH9 for the detailed review and insightful comments.
>
> ## Q1. The method does not perform well when an image consists of too much high-ferequency components.
>
> Yes. We acknowledge this as a limitation of our ES method in the discussion part (section 4), and we leave it as future work.
>
> ## Q2. Based on Figures 3, 8a-b, and 17, it seems like in practice the method is prone to higher error or failure when the changes in the PSNR curve are relatively smooth or when the noise level is low.
>
> Please refer to our General R2 (in general response) regarding the concern for our ES when facing low-level noise.
>
> ## Q3. The experiments are imbalanced and mainly focused on denoising. For example, for blind image deblurring a limited subset of the dataset and/or kernels are considered and the the results in case of the default noise levels used in the original Ren et al. (2020) solution are not provided.
>
> Thank you for pointing this out! In this revision, we have included the results on all cases for the Levin dataset, and we have also shown the results from Ren et al. (2020) in their original low-level noise. The performance of our ES method remains strong, and the conclusion does not change.
>
> ## Q4. One aspect of measuring the significance of a work is how it performs against the SOTA methods
>
> Please refer to our General R1 (in general response) on why we do not compare with state-of-the-art methods in the inverse problems that we have tested.
>
>
> ## Q5. For Table 2, it is not clear if the wall-clock time of the three ES methods should be added to DIP’s to achieve the total wall-clock time per iteration? Or that ES-WMV has some optimization resulting in lower wall-clock time per iteration when used with DIP, compared to only using DIP?
>
> Thanks for this question! Yes, the total wall-clock time per iteration should contain both the DIP and one ES method. In the revision, we have clarified this in table 2 now.
>
> ## Q6. On image deblurring: How are the overfit iteration numbers chosen?
>
> We set it as 10K, to ensure we are performing sufficient optimization. We have clarified this in Appendix A.7.12.
>
> ## Q7. The references are not written in a consistent manner. For example, few times the full name of a conference is written and at other times its abbreviation such as NeurIPS. Also, sometimes the URL and/or DOI are provided and sometimes they are not.
>
> Thanks for pointing out this and we make the references more consistent now.

---

> ### Author Response · Authors · 2022-12-05
> **We are anticipating your post-rebuttal feedback!**
>
> Dear Reviewer sEH9,
>
> We would like to sincerely thank you again for your time in reviewing our work!
>
> We understand you might be quite busy. However, as the discussion deadline is approaching, would you mind checking our response and confirming whether you have any further questions? Any further comments are discussions are welcomed!
>
> Best Regards,
>
> The authors of Paper2037

---

> ### Author Response · Authors · 2022-12-08
> **A kind reminder: Deadline of the final discussion stage (12/12) is fast approaching!**
>
> Dear reviewer sEH9,
>
> We would like to sincerely thank you again for your time and efforts for your insightful reviews!
>
> We are so sorry for keeping reminding you. We are totally okay if your final decision is to keep the score. We are writing simply to remind you that we have made the clarification and experiments as you suggested. Do they make sense to you? And possibly, do they change your mind a little bit?
>
> We value the feedback much more than a sole score. So we would really appreciate it if you could give us any feedback (like, if you think our point is not convincing, what kinds of experiments you think are missing to support the claim?). Your opinions are rather important for us to improve the work! Thanks again!
>
> Best Regards,
>
> The authors of Paper2037

---

> > ### Comment · Reviewer_sEH9 · 2022-12-10
> > **Response to authors**
> >
> > Thank you for responding to the reviewers' concerns and updating the draft, and I apologize for the delayed feedback.
> >
> > - The response regarding concerns with the performance of the proposed method in case of low level noise does not address Figure 17 (Figure 20 in the updated version) as I had pointed to in my review. To be more specific, for the top row, the detected time-steps seem to be far-off compared to the peaks which apparently are the last iteration in this case. Also, could this mean that your suggested solution to the other failure case of low level noise, reducing the learning rate, might also fail if a similarly smooth and increasing curve is resulted in such cases?
> >
> > - Regarding the failure cases in Tables 8 and 10 for the added tasks, including the related curves to visualize the behavior of the method is helpful and hopefully points to the reasons for failure.
> >
> > - I understand that the paper is not claiming SOTA results. But I still think it is helpful to include comparisons to SOTA, on each task considered. Readers need this information to better understand the importance and impact of the proposed method in the field.
> >
> > I will wait for other reviewers to respond before I update and finalize my score recommendation. Most likely it is going to be increased as many concerns are resolved.

---

> > > ### Author Response · Authors · 2022-12-12
> > > **Re: Response to authors**
> > >
> > > Dear Reviewer sEH9,
> > >
> > > Thanks for your response! We really appreciate your time and efforts! We are excited to see that many of your concerns are resolved and you plan to increase the rating!
> > >
> > > * (1) We may be able to solve the case you mentioned by adjusting the patience number. In General R2 (in general response), we acknowledge that (a) the first major valley of the VAR is always roughly aligned with their corresponding PSNR peak across all of the four DIP variants and across low- and high-level noise;  (b) our valley-detection method on certain DIP variants in low-noise regime, can be improved via finetuning the patient number.
> > >
> > > (2) More importantly, although our ES methods can help detect ES points for DIP variants, the recommended combination is still the original DIP+our ES method: (a) We justify the effectiveness of the original DIP with our ES on various tasks, including image denoising, inpainting, super-resolution, MRI reconstruction, and blind image deblurring in Section 3 and the Appendix; (b) We argue that SB will introduce several extra hyperparameters in Appendix A.6. In Figure 20, we show that the modified DIP variant—SB is very sensitive to the regularization parameter and the right regularization level often depends on the noise type/level—which are not known in practice. Inappropriate hyperparameter setting will leave the overfitting issue unsolved. So in emphasizing practicality, there are strong reasons to rule out the modified DIP variant—SB.
> > >
> > > * We are happy to address your concern!
> > >
> > > * We totally agree with your point! We will collect the SOTA results for various tasks for reference in the future modified version.

---

### Official Review · Reviewer_hubG · 2022-10-31

**Confidence:** 3
**Correctness:** 3
**Technical Novelty And Significance:** 3
**Empirical Novelty And Significance:** 3
**Recommendation:** 5

**Clarity, Quality, Novelty And Reproducibility:**

The motivation and methodology of the paper are clear. However, the experiment section is hard to follow. The method is novel, but the experiments don't show a significant improvement with the method.

**Strength And Weaknesses:**

Strength:
- The paper demonstrates the early-learning pheromone of the deep image prior both empirically and theoretically.
- A broad range of experiments are conducted, including different applications (image denoising, deblurring, MRI reconstruction)

Weakness:
- A sensitivity analysis on hyperparameters (patience number and window size) is conducted, but it's unclear which dataset it that. Figure 3  shows the pattern of training curves looks really different across different noise levels or different DIP methods/variants. Are they using the same hyperparameter?  Will the performance still be stable in that case?
- The experiments are not systematically conducted. The paper lists a lot of related works, but fails to organize the experiments in a logical way. Some of them are benchmarked on one dataset, while others are on another. What DF-STE is? Why has it only experimented on the nine-image dataset, not CBSD68?
- The improvement compared to SB is marginal. The gaps are all within one standard deviation.
- Lack of other denoising baselines, like noise2self, self2self and etc.
- How does the training loss look compared to the other training curves? Does the low variance part look flat? Can you tell anything about early-learning from the loss curve?

**Summary Of The Paper:**

The paper solves the early-stopping problem for deep image prior by tracking the variance of reconstructed images within a moving window during training. It shows that early stopping when the variance reaches the minimum results in the best deep image prior.

**Summary Of The Review:**

The paper explores an interesting early-learning phenomenon in DIP and proposes a novel early stopping method to optimize DIP. However, the experiments are not well-organized to support the claims. The organization and writing of the paper need to be improved.

---

> ### Author Response · Authors · 2022-11-19
> **Response to Reviewer hubG**
>
> We thank reviewer hubG for the detailed review and insightful comments.
>
> ## Q1. A sensitivity analysis on hyperparameters (patience number and window size) is conducted, but it's unclear which dataset it that.
>
> The analysis is done on the classical 9-image dataset discussed in our paper. Thanks for your question! We have added the setting to the ablation study (section 3.4).
>
> ## Q2. Figure 3 shows the pattern of training curves looks really different across different noise levels or different DIP methods/variants. Are they using the same hyperparameter? Will the performance still be stable in that case?
>
> The different DIP variants have different mathematical forms with different hyperparameters—we have added Appendix A.5. to describe their details. We use their default hyperparameters from their official codes (the downloading info can be found in Appendix A.7.1), and the PSNR (i.e., training) curves are expected to look different. For our ES method operating on them, we never finetune our hyperparameters, even for different noise levels.
>
> ## Q3.The experiments are not systematically conducted. The paper lists a lot of related works, but fails to organize the experiments in a logical way. Some of them are benchmarked on one dataset, while others are on another. What DF-STE is? Why has it only experimented on the nine-image dataset, not CBSD68?
>
> Thank you for raising this! The nine-image dataset has been used by all the methods under comparison in their original papers, so we have a thorough comparison using it. For CBSD68, we agree that our comparison was not systematic, and our failure to include the comparison with DF-STE was ignorance.
>
> In this revision, we have included comparisons on CBSD68 for all methods that have reported results on this dataset in their original papers.
>
> The gaps between our ES-WMV and DF-STE for all noise levels mostly come from the peak-performance between the original DIP and DF-STE---modifications in DF-STE have affected the peak performance, positively for low-level cases and negatively for high-level cases, not much from our ES method as evident from the uniformly small detection gaps reported in Table 5. Moreover, DF-STE can only handle Gaussian and Poisson noise for denoising, and the exact noise level is a required hyperparameter for their method to work.
>
> ## Q4. The improvement compared to SB is marginal. The gaps are all within one standard deviation.
>
> We agree that the improvement compared to SB is marginal. But, as we stress in the paper, SB paper consists of a combination of a DIP variant and an ES detection method. Their ES detection method only works for their DIP variant, as noted in the original SB paper "Note that the stopping criterion fails for the original deep image prior". In contrast, our ES method not only works for their DIP variant, but also for the original DIP, and other popular DIP variants. So the strength of our method is its generality, as compared to their ES method.
>
> ## Q5. Lack of other denoising baselines, like noise2self, self2self and etc.
>
> Please refer to our General R1 (in general response) on why we do not compare with state-of-the-art methods in the inverse problems that we have tested.
>
> ## Q6. How does the training loss look compared to the other training curves? Does the low variance part look flat? Can you tell anything about early-learning from the loss curve?
>
> Thank you for raising this very interesting point! We indeed have explored the possibility of using the loss for ES, but we haven’t found correlations between the trend of the loss and that of the PSNR curve. We have included visuals of the loss curves into Appendix A.7.3 in this revision.

---

> ### Author Response · Authors · 2022-12-05
> **We are anticipating your post-rebuttal feedback!**
>
> Dear Reviewer hubG,
>
> We would like to sincerely thank you again for your time in reviewing our work!
>
> We understand you might be quite busy. However, as the discussion deadline is approaching, would you mind checking our response and confirming whether you have any further questions? Any further comments are discussions are welcomed!
>
> Best Regards,
>
> The authors of Paper2037

---

> ### Author Response · Authors · 2022-12-08
> **A kind reminder: Deadline of the final discussion stage (12/12) is fast approaching!**
>
> Dear reviewer hubG,
>
> We would like to sincerely thank you again for your time and efforts for your insightful reviews!
>
> We are so sorry for keeping reminding you. We are totally okay if your final decision is to keep the score. We are writing simply to remind you that we have made the clarification and experiments as you suggested. Do they make sense to you? And possibly, do they change your mind a little bit?
>
> We value the feedback much more than a sole score. So we would really appreciate it if you could give us any feedback (like, if you think our point is not convincing, what kinds of experiments you think are missing to support the claim?). Your opinions are rather important for us to improve the work! Thanks again!
>
> Best Regards,
>
> The authors of Paper2037

---

### Official Review · Reviewer_9Jvw · 2022-10-31

**Confidence:** 4
**Correctness:** 3
**Technical Novelty And Significance:** 3
**Empirical Novelty And Significance:** Not applicable
**Recommendation:** 5

**Clarity, Quality, Novelty And Reproducibility:**

This work is clear and easy to follow with obvious novelty because it proposes a new, effective most of time, and easy enough early stopping mechanism for DIP.

**Strength And Weaknesses:**

Strength: The proposed method is simple enough and to some extent aligned with the reconstruction quality. Extensive experiments do show the superiority of the the method. And the proposed method is generally applicable to various types of noises.

Weaknesses:

1. In some cases, the PSNR gaps are still very high, up to 5dB. And according to the plots of variance curves, the proposed metho still detects early stopping time that is way off the correct time (with highest PSNR).

2. I strongly recommend the authors to provide some output images to be included in the manuscript, since PSNR is not perfectly aligned with visual quality some times (usually have some smoothening effect). Moreover, the NF-IMQ reference baselines are naturally biased towards better "quality" in their metrics instead of PSNR. Therefore, solely comparing PSNR-gaps might not be quite fair.

**Summary Of The Paper:**

This work proposes an early stopping mechanism for DIP based on the windowed moving variance of the output images. An early stop is detected when the variance does not decrease after a certain number of training steps. The proposed method is observed to have small PSNR and SSIM gaps compared to other early stopping methods and is applicable to different DIP variants.

**Summary Of The Review:**

I hold an overall positive view on this work for its technical novelty and empirical contribution.

===================================

Update after the rebuttal discussion:

After reading comments from other reviewers and the feedback from the authors, I can relate to the concerns of other reviewers that the ad-hoc manner of the proposed method, especially on the hyperparameter configuration of the patience time, creates another "early-stopping" problem. This issue is more severe considering the lack of theoretical analysis and essentially undermines the practicality of the proposed method. Co-considering this and the underwhelming performances from case to case, I decided to adjust my evaluation negatively towards this work.

---

> ### Author Response · Authors · 2022-11-19
> **Response to Reviewer 9Jvw**
>
> We thank reviewer 9Jvw for the detailed review and insightful comments.
>
> ## Q1. In some cases, the PSNR gaps are still very high, up to 5dB. And according to the plots of variance curves. The proposed method still detects early stopping time that is way off the correct time (with highest PSNR).
>
> Please refer to our General R2 (in general response) regarding the concern for our ES when facing low-level noise.
>
> ## Q2. I strongly recommend the authors to provide some output images to be included in the manuscript, since PSNR is not perfectly aligned with visual quality sometimes (usually have some smoothening effect). Moreover, the NF-IMQ reference baselines are naturally biased towards better "quality" in their metrics instead of PSNR. Therefore, solely comparing PSNR-gaps might not be quite fair.
>
> We totally agree that PSNR is not a perfect indicator of visual quality. We have tried to put more visual reconstruction results into Appendix A.7.4 given the length limit on the main paper.
>
> We agree with the potential bias, and that is why we also have SSIM gaps in the Appendix, considering that SSIM is widely believed to be better aligned with visual perception than PSNR. We have adjusted our discussion in the experiment part to emphasize this point. We do not compare the “NF-IMQ” gap, as 1) the baseline method chooses the best point according to this metric and hence leads to a zero gap, and 2)  NF-IMQ as a non-reference is much less trustable than a reference-based metric such as SSIM for similar purposes.

---

> ### Author Response · Authors · 2022-12-05
> **We are anticipating your post-rebuttal feedback!**
>
> Dear Reviewer 9Jvw,
>
> We would like to sincerely thank you again for your time in reviewing our work!
>
> We understand you might be quite busy. However, as the discussion deadline is approaching, would you mind checking our response and confirming whether you have any further questions? Any further comments are discussions are welcomed!
>
> Best Regards,
>
> The authors of Paper2037

---

> ### Author Response · Authors · 2022-12-08
> **A kind reminder: Deadline of the final discussion stage (12/12) is fast approaching!**
>
> Dear reviewer 9Jvw,
>
> We would like to sincerely thank you again for your time and efforts for your insightful reviews!
>
> We are so sorry for keeping reminding you. We are totally okay if your final decision is to keep the score. We are writing simply to remind you that we have made the clarification and experiments as you suggested. Do they make sense to you? And possibly, do they change your mind a little bit?
>
> We value the feedback much more than a sole score. So we would really appreciate it if you could give us any feedback (like, if you think our point is not convincing, what kinds of experiments you think are missing to support the claim?). Your opinions are rather important for us to improve the work! Thanks again!
>
> Best Regards,
>
> The authors of Paper2037

---

> ### Author Response · Authors · 2022-12-12
> **Response to Updated Concerns**
>
> Dear Reviewer 9Jvw,
>
> Thanks for your response and update! We really appreciate your time and efforts! We totally understand your concern about the hyperparameter configuration of the patience time for certain cases. We want to highlight three points for your consideration:
>
> * In General R2 (in general response), we acknowledge that (a) the first major valley of the VAR is always roughly aligned with their corresponding PSNR peak across all of the four DIP variants and across low- and high-level noise;  (b) our valley-detection method on DIP-TV—a variant of DIP— in low-noise regime, can be improved via finetuning the patient number. The different variants of DIP have very different forms and optimization processes, and we do not agree with the comment that the finetuning of a single hyperparameter will undermine the practicality of our method, as typical machine/deep learning methods and models involve multiple hyperparameters to finetune.
> * More importantly, although our ES methods can help detect ES points for DIP variants, the recommended combination is still the original DIP+our ES method: (a) We justify the effectiveness of the original DIP with our ES on various tasks, including image denoising, inpainting, super-resolution, MRI reconstruction, and blind image deblurring in Section 3 and the Appendix; (b) We clearly argue and show that regularization-based approaches to deal with the overfitting issue of DIP are very sensitive to the regularization parameter and the right regularization level often depends on the noise type/level—which are not known in practice. Inappropriate parameter setting will leave the overfitting issue unsolved, see, e.g., Figure 3. So in emphasizing practicality, there are strong reasons to rule out regularization-based approaches.
> * Regarding the lack of theoretical analysis, we have made it clear and also as agreed by Reviewer kTJ2, this is mostly an empirical paper that focuses on the practical effectiveness of the proposed ES method, and we have tried our best to shed light on using VAR as our detection proxy by initial but partial theoretical development.
>
> BTW, thanks to your previous second suggestion, we have added more visual results compared with baseline methods in Appendix A.7.4.  We are wondering whether you have further concerns about this point. Thanks a lot in advance!

---

### Official Review · Reviewer_cx5R · 2022-10-31

**Confidence:** 3
**Correctness:** 3
**Technical Novelty And Significance:** 3
**Empirical Novelty And Significance:** 3
**Recommendation:** 6

**Clarity, Quality, Novelty And Reproducibility:**

Clarity, quality nd reproducibility are great.
novelty is not very strong but it is an interesting topic

**Details Of Ethics Concerns:**

no concerns

**Strength And Weaknesses:**

- Strength
1. The paper is well-written with enough details
2. High-level intuition is great and easy to follow
3. Theoretical analysis is provided
4. Multiple use cases in DIP are discussed
5. The limitation is shown in different scenarios.

- Weakness
1. The theoretical analysis is not strongly related to the strategy, can you explain why the theoretical helps the early stopping criteria?
2. The experiments focus on image denoising, MRI reconstruction, and blind deblurring, why choose these three? is that possible to solve other general inverse problems, e.g., superresolution, implanting?
3. Is that early stopping DIP competitive with the SOTA approaches, e.g., diffusion-based models in these tasks?

**Summary Of The Paper:**

This paper proposes an early stopping strategy for deep image prior.  This is achieved by using an efficient ES strategy that consistently detects near-peak performance across several CI tasks and DIP variants. The paper provides high level intuition, theoretical proof and multiple use cases to demonstrate the proposed method. Also, the limitation is well discussed.

**Summary Of The Review:**

Overall, the paper is good for me but I just want to know the answer to the points listed in weakness.  Other than that, the paper is a good contribution to improving the DIP if it is competitive with the SOTA inverse problem solutions

---

> ### Author Response · Authors · 2022-11-19
> **Response to Reviewer cx5R**
>
> We thank reviewer cx5R for the detailed review and insightful comments.
>
> ## Q1. The theoretical analysis is not strongly related to the strategy, can you explain why the theoretical helps the early stopping criteria?
>
> Our theoretical analysis tries to reason why the running-variance curve has a U-shape, as shown in many figures. Our detection strategy takes advantage of the U-shape and tries to detect the valley. But we agree that our theory does not prove the valley is aligned with the performance peak as measured in PSNR, which is our empirical observation and we leave it for future theoretical development.
>
> ## Q2. The experiments focus on image denoising, MRI reconstruction, and blind deblurring, why choose these three? is that possible to solve other general inverse problems, e.g., superresolution, implanting?
>
> We have added experiments on image inpainting and superresolution into Appendix A.7.9 and A.7.10, respectively. We originally only chose denoising, MRI, and blind deblurring experiments because they represent additive, linear, and nonlinear inverse problems, respectively; indeed, our priority is to show our method can be applied to general, and especially more complicated inverse problems beyond image restoration. Super-resolution and inpainting, as classical image restoration tasks, are linear inverse problems.
>
> ## Q3. Is that early stopping DIP competitive with the SOTA approaches, e.g., diffusion-based models in these tasks?
>
> Please refer to our General R1 (in general response) on why we do not compare with state-of-the-art methods in the inverse problems that we have tested. Besides all the reasoning there, we want to add that to our best knowledge, the existing diffusion-based methods for solving inverse problems are data-driven methods.

---

> ### Author Response · Authors · 2022-12-05
> **We are anticipating your post-rebuttal feedback!**
>
> Dear Reviewer cx5R,
>
> We would like to sincerely thank you again for your time in reviewing our work!
>
> We understand you might be quite busy. However, as the discussion deadline is approaching, would you mind checking our response and confirming whether you have any further questions? Any further comments are discussions are welcomed!
>
> Best Regards,
>
> The authors of Paper2037

---

> ### Author Response · Authors · 2022-12-08
> **A kind reminder: Deadline of the final discussion stage (12/12) is fast approaching!**
>
> Dear reviewer cx5R,
>
> We would like to sincerely thank you again for your time and efforts for your insightful reviews!
>
> We are so sorry for keeping reminding you. We are totally okay if your final decision is to keep the score. We are writing simply to remind you that we have made the clarification and experiments as you suggested. Do they make sense to you? And possibly, do they change your mind a little bit?
>
> We value the feedback much more than a sole score. So we would really appreciate it if you could give us any feedback (like, if you think our point is not convincing, what kinds of experiments you think are missing to support the claim?). Your opinions are rather important for us to improve the work! Thanks again!
>
> Best Regards,
>
> The authors of Paper2037

---

### Author Response · Authors · 2022-11-19
**General Response to All Reviewers (1/2)**

We appreciate the reviewers for their thoughtful and constructive review of our manuscript. We are encouraged to see that the reviewers find our early stopping methods to be effective (All Reviewers), novel (Reviewers 9Jvw, hubG, kTJ2), and well-motivated (Reviewers cx5R, hubG, sEH9, kTJ2). All Reviewers (cx5R, 9Jvw, hubG ,sEH9, kTJ2) praise our paper for extensive experiments and most of the reviewers think our paper is well-written and easy to follow (Reviewer cx5R, 9Jvw, kTJ2). In response to feedback, we provide first a general response here to clarify several major confusing points and then individual responses below to address each reviewer’s concerns. We also update our manuscript accordingly.

## General R1. Regarding questions from Reviewers cx5R and sEH9  about comparison with state-of-the-art methods in the inverse problems that we have tested on.

Thanks for the question! Let us quickly clarify the focus and contributions of the paper. In this paper, our focus is to solve the early-learning-then-overfitting issue in DIP when used to solve visual inverse problems (VIPs), and our main contribution is proposing and validating an effective, efficient, general, and robust ES method that can locate an ES point resulting in near-peak performance with respect to the original.

It is not our intention to claim or show the original DIP and DIP+our ES (DIP+ES) methods can produce state-of-the-art (SOTA) results on any particular inverse problem. On simple inverse problems such as image denoising, image inpainting, and super-resolution, DIP is not always as competitive as SOTA data-driven methods as shown even in their original paper. While on complicated nonlinear inverse problems, DIP has obtained better results than SOTA data-driven methods, e.g., on blind image deblurring[1,2].

> Ren, D., Zhang, K., Wang, Q., Hu, Q. and Zuo, W., 2020. Neural blind deconvolution using deep priors. In Proceedings of the IEEE/CVF Conference on Computer Vision and Pattern Recognition (pp. 3341-3350).

> Zhuang, Z., Li, T., Wang, H. and Sun, J., 2022. Blind Image Deblurring with Unknown Kernel Size and Substantial Noise. arXiv preprint arXiv:2208.09483.

DIP, as a method that does not need training data, is especially useful when collecting representative training data is challenging or infeasible. So we believe it has its own niche in applications, compared to data-driven methods for VIPs. In any case, it is out of the scope of this paper to make such a conclusion as to whether DIP, or DIP+ES is the optimal method for any particular VIP. Our goal is to simply provide an effective and practical ES method for DIP.

## General R2. Regarding questions from Reviewers 9Jvw, sEH9 and kTJ2 about <strong>the concerns for our ES when facing low-level noise </strong>.

Thanks for this very insightful comment!

On detection failure in Fig 3: There are two crucial hyperparameters for our ES method: the window size (W) and the patience number (P), responsible for the VAR curve and detection of the first major valley in the VAR. From Fig 3, it is clear that for all instances, low-level and high-level noise across all of the four DIP variants, the first major valley of the VAR is always roughly aligned with their corresponding PSNR peak. So the VAR used as a reliable indicator for ES detection is valid. It is only that the detection part of our method does not work that well on DIP-TV with low-level noise. To address this issue, note that the VAR is actually observable. So one can obverse the valley-detection performance on a few instances with DIP-TV and then finetune the patience number—which determines the detection performance—to optimize the detection. For simplicity, in our paper as among the first attempts to design general ES methods for the DIP family, we have not vigorously optimized the patience number, or at a higher level, the valley-detection method itself. We leave such refinement as future work.

On failure cases in Fig 8: For the failure of GP-DIP on the “House (L)” image in Fig 8, GP-DIP has a weird multi-valley, gradual descending pattern in the VAR curve, corresponding to a multi-peak, gradual ascending pattern in the PSNR curve. The first major valley in the VAR curve is roughly aligned with the first major peak, not the final best peak, in the PSNR curve. So although our valley-detection method successfully detects the first major valley, the PSNR gap is relatively large. Overall, although our ES method works well with GP-DIP for most of the test cases, we would not recommend GP-DIP for practical use. The concern is the speed: as a method trying to mitigate the overfitting, the best reconstruction of GP-DIP tends to be around the very last iterates. The failure on the “Lena(L)" image is due to a similar multivalley pattern in the VAR curve. We have included the details in Appendix A.8.

---

> ### Author Response · Authors · 2022-11-19
> **General Response to All Reviewers (2/2)**
>
> For both cases, we observe that using smaller learning rates for GP-DIP and DD helps to smooth out their curves and mitigate the multi-valley phenomenon which likely will lead to much smaller detection gaps. We hesitate to refine in this direction, as our focus of this paper is on the ES method itself.
>
> ## General R3. According to many insightful questions from the reviewers, we have carefully revised our submission. Major modifications are as follows.
>
> * We added a section to sketch the main ideas of the DIP variants that we test in the paper in Appendix A.5.
> * We have added loss curves of the DIP process in Appendix A.7.3.
> * We added more visual results when comparing with the baseline methods in Appendix A.7.4.
> * We added comparison experiments between our ES and DF-STE on the CBSD68 dataset in Appendix A.7.5.
> * We added image inpainting experiments in Appendix A.7.9.
> * We added image super-resolution experiments in Appendix A.7.10.
> * We added more experiments comparing the two variants of our ES method, i.e., ES-WMV and ES-EMV, in Appendix A.7.11.
> * We added more blind image deblurring experiments in Appendix A.7.12.
> * We added a section to provide VAR curves for some failure cases in A.8.

---

> > ### Author Response · Authors · 2022-12-13
> > **General Response to All Reviewers after Rebuttal**
> >
> > ## General R4. Following General R2, regarding questions from Reviewer 9Jvw about the concerns for our ES when facing low-level noise.
> >
> >
> > * In General R2, we acknowledge that (a) the first major valley of the VAR is always roughly aligned with their corresponding PSNR peak across all of the four DIP variants and across low- and high-level noise; (b) our valley-detection method on DIP-TV—a variant of DIP— in low-noise regime, can be improved via finetuning the patient number. The different variants of DIP have very different forms and optimization processes, and we do not agree with the updated comment of reviewer 9Jvw that the finetuning of a single hyperparameter will undermine the practicality of our method, as typical machine/deep learning methods and models involve multiple hyperparameters to finetune.
> > * More importantly, **although our ES methods can help detect ES points for DIP variants, the recommended combination is still the original DIP+our ES method**: (a) We justify the effectiveness of the original DIP with our ES on various tasks, including image denoising, inpainting, super-resolution, MRI reconstruction, and blind image deblurring in Section 3 and the Appendix; (b) We clearly argue and show that regularization-based approaches to deal with the overfitting issue of DIP are very sensitive to the regularization parameter and the right regularization level often depends on the noise type/level—which are not known in practice. Inappropriate parameter setting will leave the overfitting issue unsolved, see, e.g., Figure 3. So in emphasizing practicality, there are strong reasons to rule out regularization-based approaches.

---

### Public Comment · ~Hao_Zhou15 · 2023-02-15
**Contradiction between the main idea of the literature and result from experiment A.7.3**

In Fig.2 you said, "Our method relies on the VAR curve, whose valley is often well aligned with the MSE valley, to detect the MSE valley—that corresponds to the PSNR peak."

However, in the results of experiment A.7.3, you said, "We also explore the possibility of using the loss for ES here, but we fail to find correlations between the trend of the loss and that of the PSNR curve."

So basically, your analysis of experimental results diminishes your idea of "finding the MSE valley—that corresponds to the PSNR peak".

Could you explain the contradiction?

Thank you.

---

> ### Author Response · Authors · 2023-02-15
> **Response to "Contradiction between the main idea of the literature and result from experiment A.7.3"**
>
> Hi Hao,
>
> Thanks for your question! In Figure 2, the MSE curve is commonly used to determine when to stop DIP, but it requires knowledge of the ground truth image, which is usually not available in practice. Hence, this is an important practical issue for DIP. In this paper, we find that the valley of the VAR curve is often well aligned with the MSE valley and then propose our stopping criterion for DIP.
>
> In A.7.3, The loss curve, which measures the difference between the reconstructed and degraded images, is also explored as a potential stopping criterion, but no correlation is found between its trend and that of the PSNR or MSE curve.
>
>
> Best Regards,
>
> The authors of Paper2037

---

### Decision · Program_Chairs · 2023-01-20

**Decision:**

Reject

**Justification For Why Not Higher Score:**

While the theoretical contribution isn't major, the empirical benefit is sometimes unobvious either.

**Justification For Why Not Lower Score:**

N/A

**Metareview: Summary, Strengths And Weaknesses:**

This paper proposed an empirical technique to detect peak DIP performance without a clean reference image. The proposed strategy utilizes the running variance of past reconstruction iterates to determine when to terminate training. The observations are certainly interesting. The paper is very well-written and easy to follow.

Two factors hold this work from being more positively assessed. On one hand, the method's performance is underwhelming sometimes (the gap can be really big, even not compared to the strongest SOTAs). The authors explained they "have not vigorously optimized the patience number, or at a higher level, the valley-detection method itself". But that brought up further concerns about how ad-hoc and sensitive their method (detection part) could perform in the presence of real and varying noise. On the other hand, as the authors also agreed in their rebuttal, the theoretical contribution of this paper is rather thin and incomplete. It is also not strongly related to the algorithm.